# DiLu🐎: A Knowledge-Driven Approach to Autonomous Driving with Large Language Models

**Licheng Wen**[1,*]**, Daocheng Fu**[1,*]**, Xin Li**[2,1,*]**, Xinyu Cai**[1]**, Tao Ma**[1,3]

**Pinlong Cai**[1]**, Min Dou**[1]**, Botian Shi**[1,†]**, Liang He**[2]**, Yu Qiao**[1]

[1] Shanghai Artificial Intelligence Laboratory

[2] East China Normal University,  [3] The Chinese University of Hong Kong

[*] Equal contributions;  [†] Corresponding author `shibotian@pjlab.org.cn`

## Abstract

Recent advancements in autonomous driving have relied on data-driven approaches, which are widely adopted but face challenges including dataset bias, overfitting, and uninterpretability. Drawing inspiration from the knowledge-driven nature of human driving, we explore the question of how to instill similar capabilities into autonomous driving systems and summarize a paradigm that integrates an interactive environment, a driver agent, as well as a memory component to address this question. Leveraging large language models (LLMs) with emergent abilities, we propose the DiLu framework, which combines a Reasoning and a Reflection module to enable the system to perform decision-making based on common-sense knowledge and evolve continuously. Extensive experiments prove DiLu's capability to accumulate experience and demonstrate a significant advantage in generalization ability over reinforcement learning-based methods. Moreover, DiLu is able to directly acquire experiences from real-world datasets which highlights its potential to be deployed on practical autonomous driving systems. To the best of our knowledge, we are the first to leverage knowledge-driven capability in decision-making for autonomous vehicles. Through the proposed DiLu framework, LLM is strengthened to apply knowledge and to reason causally in the autonomous driving domain.

Project page: `https://pjlab-adg.github.io/DiLu/`

## 1 Introduction

Autonomous driving has witnessed remarkable advancements in recent years, propelled by the data-driven manner (Bogdoll et al., 2021; Chen et al., 2023a;b). These data-driven algorithms strive to capture and model the underlying distributions of the accumulated data (Bolte et al., 2019; Zhou & Beyerer, 2023), but they always encounter challenges such as dataset bias, overfitting, and uninterpretability (Codevilla et al., 2019; Jin et al., 2023). Exploring methods to mitigate these challenges could lead to a deeper understanding of driving scenarios and more rational decision-making, potentially enhancing the performance of autonomous driving systems.

Drawing inspiration from the profound question posed by LeCun (2022): *"Why can an adolescent learn to drive a car in about 20 hours of practice and know how to act in many situations he/she has never encountered before?"*, we explore the core principles that underlie human driving skills and raise a pivotal distinction: human driving is fundamentally knowledge-driven, as opposed to data-driven. For example, when faced with a situation

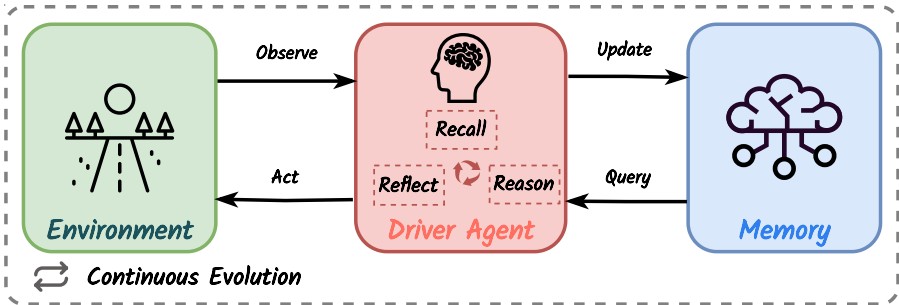

Figure 1: The knowledge-driven paradigm for autonomous driving system, including an interactive environment, a driver agent with recall, reasoning and reflection abilities, along with an independent memory module. Driver agent continuously evolves to observe the environment, query, update experiences from the memory module, and make decisions to control the ego vehicle.

where the truck ahead is in danger of losing its cargo, humans can rely on common sense and explainable reasoning to ensure a safe distance is maintained between vehicles. Conversely, data-driven methods rely on a large quantity of similar data to fit this scenario which lacks environment comprehension and limits generalization ability (Heidecker et al., 2021; Chen et al., 2022; Wen et al., 2023). Furthermore, this task requires significant human labor and financial resources to collect and annotate driving data to handle varied real-world scenarios. This observation catalyzes a fundamental question: **How can we instill such knowledge-driven capabilities of human drivers into an autonomous driving system?**

Recent advancements in large language models (LLMs) with emergent abilities offer an ideal embodiment of human knowledge, providing valuable insights toward addressing this question. LLMs possess exceptional human-level abilities and show strong abilities in robotics manipulation (Driess et al., 2023a; Huang et al., 2023a;b), multi-modal understanding (Gao et al., 2023) and lifelong skill learning (Wang et al., 2023; Zhu et al., 2023b). However, just as humans may need 20 hours of practice to learn to drive, LLMs cannot successfully perform the driving task without any experience or guidance. Through these analyses, we summarize the knowledge-driven paradigm for autonomous driving systems, as illustrated in Figure 1, including three components: (1) an environment with which an agent can interact; (2) a driver agent with recall, reasoning, and reflection abilities; (3) a memory component to persist experiences. In continuous evolution, the driver agent observes the environment, queries, and updates experiences from memory and performs decision-making.

Following the paradigm above, we design a novel framework named DiLu as illustrated in Figure 2. Specifically, the driver agent utilizes the **Reasoning Module** to query experiences from the **Memory Module** and leverage the common-sense knowledge of the LLM to generate decisions based on current scenarios. It then employs the **Reflection Module** to identify safe and unsafe decisions produced by the Reasoning Module, subsequently refining them into correct decisions using the knowledge embedded in the LLM. These safe or revised decisions are then updated into the Memory Module.

Extensive experiments demonstrate that the proposed framework DiLu can leverage LLM to make proper decisions for the autonomous driving system. We design a closed-loop driving environment and prove that DiLu can perform better and better with the experience accumulated in the memory module. Remarkably, with only 40 memory items, DiLu achieves comparable performance to the reinforcement learning (RL) based methods that have extensively trained over 600,000 episodes, but with a much stronger generalization ability to diverse scenarios. Moreover, DiLu's ability to directly acquire experiences from real-world datasets highlights its potential to be deployed on practical autonomous driving systems.

The contributions of our work are summarized as follows:

- To the best of our knowledge, we are the first to leverage knowledge-driven capability in decision-making for autonomous vehicles from the perspective of how humans drive. We summarize the knowledge-driven paradigm that involves an interactive environment, a driver agent, as well as a memory component.

- We propose a novel framework called DiLu which implements the above paradigm to address the closed-loop driving tasks. The framework incorporates a Memory Module to record the experiences, and leverages LLM to facilitate reasoning and reflection processes.

- Extensive experimental results highlight DiLu's capability to continuously accumulate experience by interacting with the environment. Moreover, DiLu exhibits stronger generalization ability than RL-based methods and demonstrates the potential to be applied in practical autonomous driving systems.

## 2 RELATED WORKS

### 2.1 ADVANCEMENTS IN LARGE LANGUAGE MODELS

Large language models (LLMs) are the category of Transformer-based language models that are characterized by having an enormous number of parameters, typically numbering in the hundreds of billions or even more. These models are trained on massive text datasets, enabling them to understand natural language and perform a wide range of complex tasks, primarily through text generation and comprehension (Zhao et al., 2023). Some well-known examples of LLMs include GPT-3 (Brown et al., 2020), PaLM (Chowdhery et al., 2022), and LLaMA (Touvron et al., 2023), GPT-4 (Achiam et al., 2023). The emergent abilities of LLMs are one of the most significant characteristics that distinguish them from smaller language models. Specifically, in-context learning (ICL) (Brown et al., 2020), instruction following (Ouyang et al., 2022; Wei et al., 2021) and reasoning with chain-of-thought (CoT) (Wei et al., 2022) are three typical emergent abilities for LLMs. OpenAI's pursuit of LLMs has led to the achievement of two remarkable milestones: ChatGPT (OpenAI, 2023) and GPT-4 (Achiam et al., 2023). These two milestones signify significant advancements in LLMs' capabilities, particularly in natural language understanding and generation. Notably, recent developments in large LLMs have showcased human-like intelligence and hold the potential to propel us closer to the realm of Artificial General Intelligence (AGI) (Zhao et al., 2023; Zhu et al., 2023a).

### 2.2 ADVANCED TASKS BASED ON LARGE LANGUAGE MODEL

Owing to the superior capability of common-sense knowledge embedded in LLMs, they are widely adopted for diverse tasks (Sammani et al., 2022; Bubeck et al., 2023; Schick et al., 2023). Furthermore, there has emerged a flourishing research area that leverages LLMs to create autonomous agents endowed with human-like capabilities (Chowdhery et al., 2022; Yao et al., 2022; Park et al., 2023; Fu et al., 2024; Zhu et al., 2023b; Li et al., 2023). In particular, LLMs are shown to possess a wealth of actionable knowledge that can be extracted for robot manipulation in the form of reasoning and planning. For instance, (Driess et al., 2023b) proposed embodied language models that directly integrate real-world continuous sensor data into language models, establishing a direct link between words and perceptual information. Voyager (Wang et al., 2023) introduces lifelong learning through the incorporation of prompting mechanisms, a skill library, and self-verification. These three modules are all grounded by LLM and empower the agent to learn more sophisticated behaviors. Similarly, Voxposer(Huang et al., 2023b) leveraged LLMs to generate robot trajectories for a wide range of manipulation tasks, guided by open-ended instructions and objects. Simultaneously, motivated by insights from AGI and the principles of embodied AI (Pfeifer & Iida, 2004; Duan et al., 2022), the field of autonomous driving is also undergoing a profound transformation. An open-loop driving commentator LINGO-1 is proposed by Wayve (2023) which combines vision, language and action to enhance how to interpret and train the driving models. LLMs have demonstrated remarkable human-level capabilities across various domains, but we observe that they lack the inherent ability to engage with and comprehend the complex driving environment as humans. In contrast, autonomous vehicles depend on systems that can actively interact with and understand the driving environment. To bridge this gap, we propose a novel knowledge-driven autonomous driving paradigm and DiLu framework that enables LLMs to comprehend driving environments and drive by incorporating human knowledge.

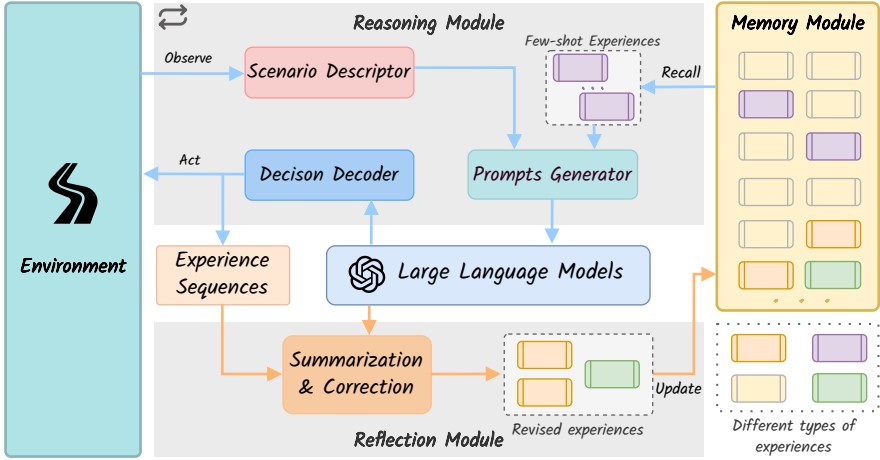

Figure 2: The framework of DiLu. It consists of four modules: Environment, Reasoning, Reflection, and Memory. In DiLu, the Reasoning module can observe the environment, generate prompts by combining scenario descriptions and experiences in the Memory module, and decode responses from the LLM to finish decision-making. Concurrently, the Reflection module evaluates these decisions, identifies the unsafe decision to the experiences, and finally updates the revised experiences into the Memory module.

## 3 METHODOLOGY

### 3.1 OVERVIEW

Based on the knowledge-driven paradigm for autonomous driving systems introduced previously, we propose a practical framework called DiLu, as illustrated in Figure 2. DiLu consists of four core modules: Environment, Reasoning, Reflection, and Memory. In particular, the Reasoning module begins by observing the environment and obtaining descriptions of the current scenario. Concurrently, a prompt generator is employed to combine this scenario description with the few-shot experiences of similar situations, which retrieved from the Memory module. These prompts are then fed into an out-of-the-box Large Language Model (LLM), and the decision decoder make an action by decoding LLM's response.

This process is iterated within the Reasoning module, resulting in time-series decision sequences. Subsequently, we employ the Reflection module to assess past decision sequences, categorizing them as either safe or unsafe. The unsafe decisions are revised and these refined decisions are finally updated back into the Memory module. Detailed implementations of the Memory, Reasoning, and Reflection modules will be elaborated in the following sections.

### 3.2 MEMORY MODULE

Without few-shot experiences, the out-of-the-box LLMs fail to perform precise reasoning when tackling the complex closed-loop driving tasks. Therefore, we employ a Memory module to store the experiences from past driving scenarios, which include the decision prompts, reasoning processes, and other valuable information. The memory stored in the memory module consists of two parts: scene descriptions and corresponding reasoning processes. The scene description provides a detailed account of the situation, serving as the key to the memory module for retrieving similar memories. The reasoning process, on the other hand, records the appropriate method for handling the situation. This is the value of the memory, guiding the agent towards the correct driving logic. The memory module is constructed in three stages: initialization, memory recall, and memory storage.

**Initialization**: The Initialization of memory module is akin to a human attending driving school before hitting the road. We select a few scenarios and manually outline the correct reasoning and decision-making processes for these situations to form the initial memory. These memories instruct the agent on the correct decision-making process for driving.

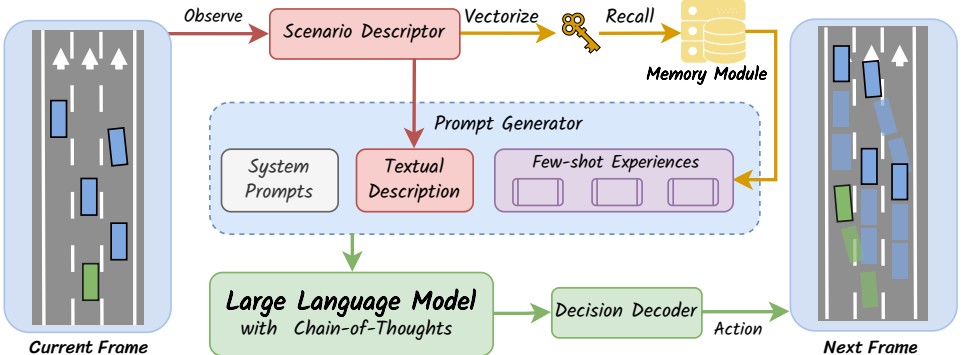

Figure 3: Reasoning module. We leverage the LLM's common-sense knowledge and query the experiences from Memory module to make decisions based on the scenario observation.

**Memory recall**: At each decision frame, the agent receives a textual description of the driving scenario. Before making a decision, the current driving scenario is embedded into a vector, which serves as the memory key. This key is then clustered and searched to find the closest scenarios (Johnson et al., 2019) in the memory module and their corresponding reasoning processes, or memories. These recalled memories are provided to the agent in a few-shot format to assist in making accurate reasoning and decisions for the current scenario.

**Memory storage**: As the agent makes correct reasoning and decisions, or reflects on the correct reasoning process, it gains driving experience. We embed the scene description into a key, pair it with the reasoning process to form and store memory in the memory module.

### 3.3 Reasoning Module

In the Reasoning module, we utilize the experiences derived from the Memory module and the common-sense knowledge of the LLM to perform decision-making for the current traffic scenario. Specifically, the reasoning procedure is illustrated in Figure 3, including the following procedures: (1) encode the scenario by a descriptor; (2) recall several experience from the Memory module; (3) generate the prompt; (4) feed the prompt into the LLM; (5) decode the action from the LLM's response. The detailed prompt design can be found in Appendix A.2.

**Encode the scenario by a descriptor:** To facilitate DiLu's understanding of the current traffic conditions, the scenario descriptor transcribes the present scenario data into descriptive text. The scenario descriptor follows a standard sentence structure and utilizes natural language to offer a comprehensive depiction of the ongoing driving scenario. This description contains static road details as well as dynamic information regarding the ego vehicle and surrounding vehicles within the scenario. These generated descriptions are then used as input for the prompt generator and as keys to acquire relevant few-shot experiences from the Memory module.

**Recall several experience from the Memory module:** During the reasoning process, the description of the current driving scenario is also embedded into a vector, as illustrated in Figure 3. This vector is then used to initiate a similarity query within the Memory module, searching for the top k similar situations. The resulting paired scene descriptions and reasoning procedures assemble as few-shot experiences, which are then integrated into the prompt generator.

**Generate the prompt:** As depicted in the blue dashed box in Figure 3, the prompts for each frame consist of three key components: system prompts, textual descriptions from scenario descriptor, and the few-shot experience. Within the system prompts, we offer a concise overview of the closed-loop driving task, which includes an introduction to the content and format of task inputs and outputs, along with constraints governing the reasoning process. In each decision frame, we construct tailored prompts based on the current driving

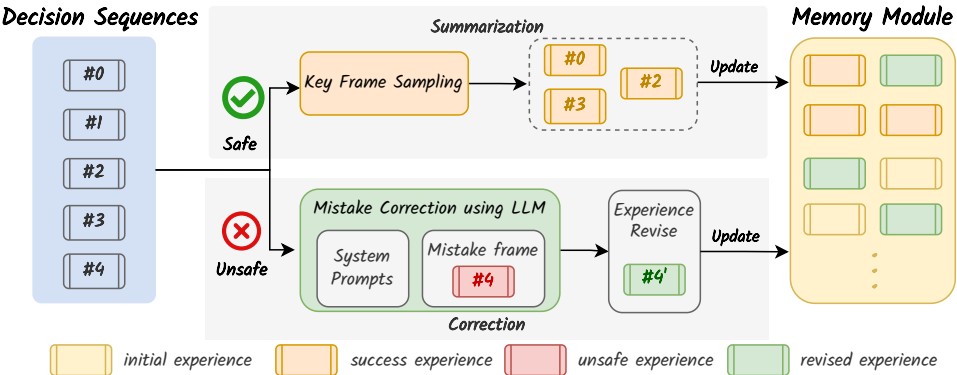

Figure 4: Reflection module. The Reflection module takes recorded decisions from closed-loop driving tasks as input, it utilizes a summarization and correction module to identify safe and unsafe decisions, subsequently revising them into correct decisions through the human knowledge embedded in LLM. Finally, these safe or revised decisions are updated into Memory module.

scenario. These prompts are subsequently employed by LLM to engage in reasoning and determine the appropriate action for the current frame.

**Feed the prompt into the LLM:** Since the closed-loop driving task requires a complex reasoning process to make proper decisions, we employ the Chain-of-Thought (CoT) prompting techniques introduced by Wei et al. (2022). These techniques require LLM to generate a sequence of sentences that describe the step-by-step reasoning logic, ultimately leading to the final decision. This approach is adopted due to the inherent complexity and variability of driving scenarios, which may lead to hallucinations if the language model directly produces decision results. Also, the generated reasoning process facilitates further refinement and modification of incorrect decisions by the Reflection module in Section 3.4.

**Decode the action from the LLM's response:** After feeding the prompt into the LLM, we decode the final decision in action decoder. The action decoder translates LLM's decision outcomes into actions for the ego vehicle and provides feedback to the environment. By repeating the procedures above, we establish a closed-loop decision-making system.

## 3.4 REFLECTION MODULE

In the Reasoning module, we use the LLM to undertake closed-loop driving tasks with the support of the proposed Memory module. As a next step we hope to accumulate valuable experiences and enrich the Memory module upon the conclusion of a driving session. To achieve this goal, we propose the Reflection module in DiLu, which continuously learns from past driving experiences. DiLu can progressively improve its performance through the Reflection module, similar to the progression of a novice becoming an experienced driver.

The Reflection module is illustrated in Figure 4. During the closed-loop driving task, we record the prompts used as input based on the driving scenario and the corresponding decisions generated by the LLM for each decision frame. Once a driving session concludes, we obtain a decision sequence, e.g., 5 decision frames from 0 to 4 in Figure 4. When the session ends without any collisions or hazardous incidents, indicating a successful session, DiLu proceeds to sample several key decision frames from the sequence. These frames then directly become part of the historical driving experience and enrich the Memory module.

On the contrary, if the current session is terminated due to hazardous situations such as collisions with other vehicles, this indicates that the driver agent has made inaccurate decisions. It is crucial for the system to rectify the unsafe decision made by the Reasoning module. Thanks to the interpretable chain-of-thoughts responses, we can easily find the causes of dangerous situations. Certainly, we can ask a human expert to complete such an error correction process. However, our goal is to make the autonomous driving system learn

from mistakes on its own. We discover that LLM can effectively act as a mistake rectifier. Our approach is to use the driving scenarios in which incorrect decisions occurred, together with the original reasoning output, as prompts for LLM. We instruct LLM to pinpoint the reasons behind the incorrect decision and provide the correct one. We also ask LLM to propose strategies in order to avoid similar errors in the future. Finally, the correct reasoning process and the revised decision learned from the mistakes are retained in Memory module.

# 4 EXPERIMENTS

In our experimental setup, we utilize the well-established Highway-env as our simulation environment, which is a widely used platform in the fields of autonomous driving and tactical decision-making (Leurent, 2018). This environment provides several driving models and offers a realistic multi-vehicle interaction environment. In addition, the density of vehicles and the number of lanes in the environment can be freely adjusted. The detailed setup of DiLu framework and Highway-env are described in Appendix A.1. The demonstration video of DiLu can be found in the project page: `https://pjlab-adg.github.io/DiLu/`.

## 4.1 THE VALIDATION OF THE DILU FRAMEWORK

In this section, we primarily focus on validating the effectiveness of the DiLu framework, in particular the reasoning and reflection process with or without the Memory module. The DiLu framework without the Memory module is referred to as the 0-shot baseline, and we conduct comparative experiments with 1-shot, 3-shots, and 5-shots experiences to demonstrate the necessity of experience accumulation. The initial Memory module contains 5 human-crafted experiences. These experiences are then used in different few-shot settings for reasoning and reflection, allowing for continuous accumulation and updating of experiences. We conduct comparative experiments when the Memory module has 5, 20, and 40 experiences, respectively. Each setting is repeated 10 times with different seeds. The results are shown as a box plot in Figure 5 (a). Success Steps (SS) is the number of consecutive frames without collision, an SS of 30 means that the ego car has completed the driving task. We found that as the number of experiences in the Memory module increases, the performance of the DiLu framework improves in all few-shot settings. Notably, the 5-shots setting successfully completes the closed-loop driving task in the majority of cases in scenarios with 20 experiences. Furthermore, when the Memory module is equipped with 40 experiences, all trials achieve a median SS of over 25. In comparison, when the framework lacks the Memory module and runs a 0-shot experiment, no tasks are successfully conducted and the median SS is below 5. This indicates that LLM cannot directly perform the closed-loop driving tasks without any adaptation.

In addition, we observed that for a fixed number of memory items, the performance of the framework improved as the number of few-shot experiences increased. Specifically, with 40 memory items, the 5-shot framework successfully passed almost all tests, while the median SS for the 3-shots and 1-shot frameworks were 27 and 25, respectively. This can be attributed to the fact that a higher number of few-shots includes a diverse range of experiences in similar driving scenarios. When fed into the LLM as prompts, these allow the LLM to draw on a wider range of information and decision strategies, thereby facilitating more rational decisions. Several detailed case studies can be found in Appendix A.3.

## 4.2 COMPARISON WITH REINFORCEMENT LEARNING METHOD

We conducted comparative experiments between DiLu and the SOTA reinforcement learning (RL) method in Highway-env, GRAD (Graph Representation for Autonomous Driving) (Xi & Sukthankar, 2022). GRAD is an RL-based approach specifically designed for autonomous driving scenarios, it generates a global scene representation that includes estimated future trajectories of other vehicles. We trained GRAD under the setting of `lane-4-density-2`, which means the 4-lane motorway scenarios with vehicle density of 2.0. The training details are given in Appendix A.4 and A.5. As a comparison, DiLu used 40 experiences purely obtained from `lane-4-density-2` setting. We defined the success rate (SR) as driving

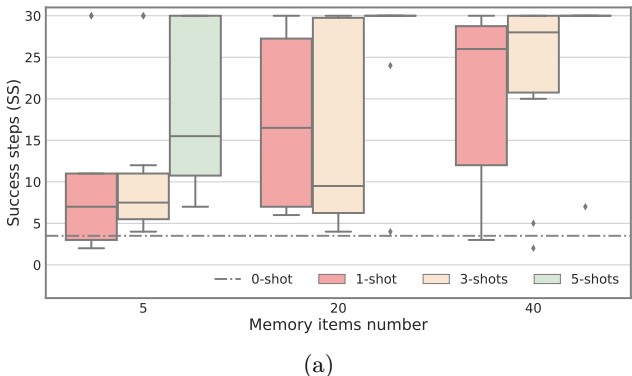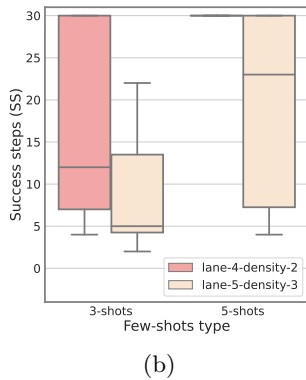

(a)                         (b)

Figure 5: (a) Quantitative experiments with different experiences in Memory module and different few-shot numbers. Notably, the 5-shots setting achieve a maximum(30) simulation steps with both 20 and 40 memory items. (b) Generalizability experiment on environment with different traffic density using 20 memory items.

without any collision for 30 decision frames and then conducted experiments under three environment settings: `lane-4-density-2`, `lane-5-density-2.5`, and `lane-5-density-3`. Each setup includes 10 test scenarios with different seeds.

The results are illustrated in Figure 6 (a). Firstly, in the `lane-4-density-2` setting, DiLu uses **only 40 experience** in the Memory module to achieve 70% SR while the GRAD converges to 69% SR after 600,000 training episodes. We found that many failures from GRAD are due to the inability to brake in time, resulting in collisions with the vehicle ahead. This is because reinforcement learning methods tend to fit the environment and fail to take human-like driving knowledge into consideration. Secondly, we migrate DiLu and GRAD optimized on `lane-4-density-2` to `lane-5-density-2.5` and `lane-5-density-3` settings. We observe that both methods suffer varying degrees of performance degradation in the migrated environments, as the number of lanes changes and traffic density increases. However, in the most complex `lane-5-density-3` environment, DiLu still maintains a 35% SR without extra optimization, while the GRAD has an 85% performance drop. DiLu accumulated experience in one environment can be generalized into another one, while RL method tends to overfit the training environment.

### 4.3 EXPERIMENTS ON GENERALIZATION AND TRANSFORMATION

Data-driven approaches often overfit the training environment, while human knowledge should be domain-agnostic and can be generalized to different environments. We perform several experiments to evaluate DiLu's generalization and transformation abilities.

**Generalization ability in different environments.** We verify whether the experiences obtained in DiLu's Memory module have generalization capabilities. More formally, we used 20 experiences obtained from the `lane-4-density-2` environment, and conducted experiments in the `lane-5-density-3` setting, testing the closed-loop performance of 3-shot and 5-shot respectively. The experimental results are shown in Figure 5 (b). As we can see, the 3-shot version of DiLu achieves 13 median SS on the setting of `lane-4-density-2` while decreasing to 5 media SS under the `lane-5-density-3` setting. But in the 5-shot version, we achieve 30→23 median SS degradation in the same situation. This suggests that the ability to generalize is better with more few-shot experience fed into the LLM.

**Transformation ability using real-world dataset.** Since DiLu's Memory module stores experiences in the form of natural language text, it contains environment-agnostic knowledge that can be readily transferred to different environments. To illustrate this capability, we created two Memory modules, each containing 20 experiences extracted from two distinct sources: (1) Highway-env and (2) CitySim, a dataset comprising real-world vehicle trajectory data (Zheng et al., 2023). We subsequently evaluated these modules on the `lane-4-density-2` and `lane-5-density-3` scenarios within the Highway-env environment. The experimental results are presented in Figure 6 (b). In comparison to a system without any prior experiences (depicted by the gray dash-dot line), CitySim-contributed knowledge

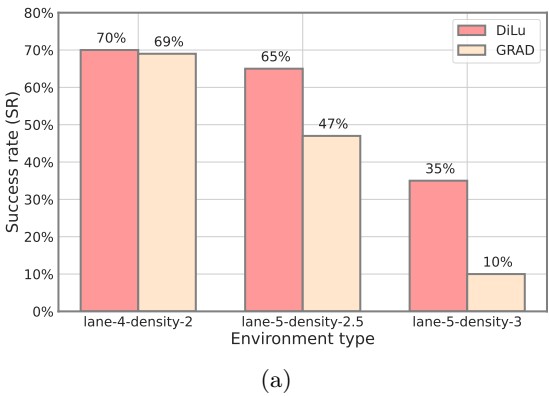 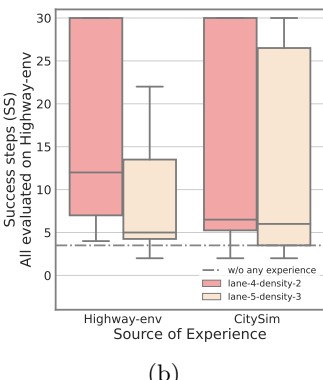

(a)                                                         (b)

Figure 6: (a) Performance comparison with GRAD in different types of motorway environments. Both two methods are **only** optimized on `lane-4-density-2` settings and evaluate on `lane-4-density-2`, `lane-5-density-2.5` and `lane-5-density-3` respectively. (b) Experiments with experience from different domains.

enhanced DiLu's performance. Also, when transferring experiences to a more congested environment (from `lane-4-density-2` to `lane-5-density-3`), the memory derived from real-world data exhibited superior robustness compared to the memory accumulated solely within the simulation domain.

### 4.4 Effectiveness of two memory types in the Reflection module

In this section, we explore the significance of incorporating successful experiences and revised unsafe experiences in the Reflection module through an ablation study. We adopt the Memory module containing 20 initial experiences and observe the performance change during the new memory accumulation. The results are shown in Table 1. The `baseline` indicates the system using the Memory module with 20

Table 1: Effectiveness of two memory types in the Reflection module on Success Steps. MIN, Q1, Median, Q3, MAX means the quartile statistical evaluation.

| Methods | MIN | Q1 | Median | Q3 | MAX |
|---|---|---|---|---|---|
| `baseline` | 2.0 | 3.0 | 10.0 | 22.0 | 30.0 |
| `+success_memory` | 3.0 | 7.0 | 24.5 | 29.3 | 30.0 |
| `+correction_memory` | 4.0 | 7.75 | 21.5 | 30.0 | 30.0 |
| `+both_type_memory` | **5.0** | **7.8** | **24.5** | **30.0** | **30.0** |

initial experiences. Then we add 12 success memories and 6 correction memories into the baseline successively. In the experiments, the median success step of the `baseline` is only 10. However, after updating with new experiences, all two methods achieve a median success step of over 20, and the method with both types of experiences shows higher success steps on all statistical measures. Therefore, it is both reasonable and effective to add two different types of experiences during reflection.

## 5 Conclusion

In this paper, we explore the realm of instilling human-level knowledge into autonomous driving systems. We summarize a knowledge-driven paradigm and propose the DiLu framework, which includes a memory module for recording past driving experiences and an agent equipped with Reasoning and Reflection modules. Extensive experimental results showcase DiLu's capability to continuously accumulate experience and exhibit its strong generalization ability compared to the SOTA RL-based method. Moreover, DiLu's ability to directly acquire experiences from real-world datasets highlights its potential to be deployed on practical autonomous driving systems. The DiLu framework, while effective, is not without limitations. Presently, it experiences a decision-making latency of 5-10 seconds, encompassing LLM inference and API response times. Additionally, it does not completely eradicate hallucinations generated by LLMs. Future improvements could capitalize on recent advances in LLM compression and optimization, aiming to enhance both efficiency and effectiveness.

ACKNOWLEDGMENTS

The research was supported by Shanghai Artificial Intelligence Laboratory, the National Key R&D Program of China (Grant No. 2022ZD0160104) and the Science and Technology Commission of Shanghai Municipality (Grant Nos. 22DZ1100102 and 23YF1462900).

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

## A APPENDIX

### A.1 DETAILED SETUP OF THE EXPERIMENT

Within our DiLu framework, the large language model we adopt is the GPT family developed by OpenAI. GPT-3.5 (OpenAI, 2023) is used in the Reasoning module of the framework, which is responsible for making reasonable decisions for the ego vehicle. GPT-4 is used in the Reflection module since it has demonstrated significantly improved self-repair and fact-checking capabilities compared to GPT-3.5 (Bubeck et al., 2023; Olausson et al., 2023). To serve as the Memory module in the DiLu framework, we adopt Chroma[1], an open-source embedding vector database. The scenario descriptions are transformed into vectors using the `text-embedding-ada-002` model of OpenAI.

In terms of the setup for highway-env, we directly obtain vehicle information from the underlying simulation and input it into the scenario descriptor. This information only includes each vehicle's position, speed, and acceleration data in the current frame, without any decision intent or potential risk information, as shown in Figure 7. Meta-actions are used as the decision output in our experiments, which include five discrete actions to control the ego vehicle: acceleration, maintaining speed, deceleration, and lane changes to the left and right. For each closed-loop driving task, we define a successful completion time of 30 seconds, with a decision-making frequency of 1Hz. This means that if the ego vehicle can navigate through traffic at a reasonable speed and without collisions for 30 seconds, we consider the task to be successfully completed. Unless otherwise stated, our experimental environment is a four-lane motorway with a vehicle density of 2.0, representing scenarios with relatively high traffic density and complexity. All other settings follow the simulator's default configurations.

### A.2 PROMPTS EXAMPLE

In this section, we detail the specific prompts design in the reasoning and reflection modules.

**Reasoning prompts** As mentioned in the article, the prompts for the reasoning module primarily consist of three parts: system prompts, scenario description, and few-shot experience. Specifically, as shown in Figure 7, the system prompts section is entirely fixed and mainly includes an introduction to the closed-loop driving task, instructions for input and output, and formatting requirements for LLM responses. Most of the scenario description is fixed, but three parts are directly related to the scenario and are dynamically generated based on the current decision frame. The driving scenario description contains information about the positions, speeds, and accelerations of the ego vehicle and surrounding key vehicles. It's important to note that we only embed the text of the driving scenario description into vectors and use it as a query input to the memory module. Available action includes all meta-actions. The driving intention can be input by humans to modify the vehicle's behavior, with the default intention being: "Your driving intention is to drive safely and avoid collisions."

As for the few-shot experience, it is entirely obtained from the memory module. Each experience consists of a human-LLM dialogue pair, where the human question includes the scenario description at that decision frame, and the LLM response represents the correct (or correct after revised) reasoning and decision made by the driver agent. The extracted experiences are directly utilized with a few-shot prompting technique to input into the large language model, enabling in-context learning. Figure 8 demonstrates the results of a 3-shot experience query, which includes two "keep speed" decisions and one "decelerate" decision. It's important to note that consistency in decisions is not a requirement within the few-shot experience.

---

[1]`https://github.com/chroma-core/chroma`

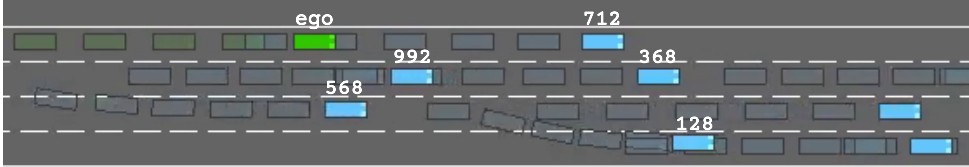

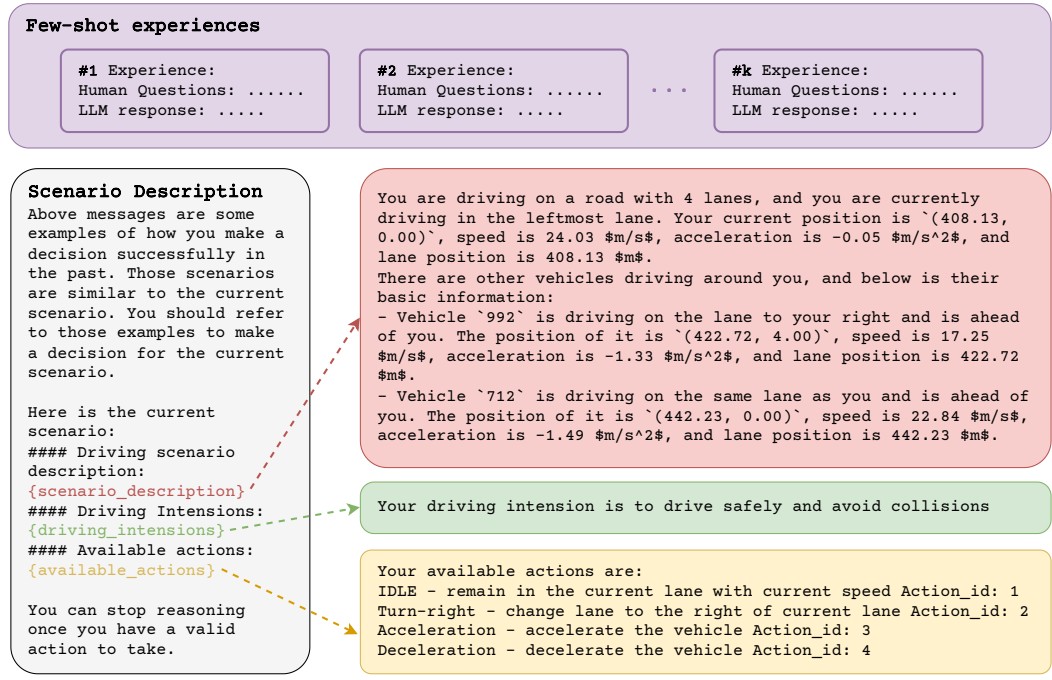

Figure 7: The prompts template for reasoning module. The prompts in the grey box are fixed, while the prompts in the coloured box are different depending on the current scenario. The few-shot experiences are detailed demonstrated in Figure 8.

**Human Question:**
Here is the #1 scenario:
#### Driving scenario description: Description of the scene, omitted here...
#### Driving Intensions: Description of the intension, omitted here...
#### Available actions: Description of the available actions, omitted here...
You can stop reasoning once you have a valid action to take.

**LLM response:**
I have 4 actions to choose from. Now, I would like to know which action is possible.
I should first check if I can accelerate, then idle, finally decelerate. I can also try to change lanes but with caution and not too frequently.
- I want to know if I can accelerate, so I need to observe the car in front of me on the current lane, which is car `176`. The distance between me and car `176` is 464.26 - 432.15 = 32.11 m, and the difference in speed is 20.98 - 24.00 = -3.02 m/s. Car `176` is traveling 32.11 m ahead of me and its speed is 3.02 m/s slower than mine. This distance is safe and my speed is already higher than the front car on the current lane. So I should not accelerate.
- Since I cannot accelerate, I want to know if I can maintain my current speed. I need to observe the car in front of me on the current lane, which is car `176`. Due to above calculation, the distance between me and car `176` is 32.11 m, and the difference in speed is -3.02 m/s. The distance 32.11 m is larger than my preferred car following distance 30 m, and my speed is higher than the front car on the current lane. So I can maintain my current speed.
Great, I can make my decision now. Decision: IDLE
Response to user:#### 1

**#1 Experience**

**Human Question:**
Here is the #2 scenario:
#### Driving scenario description: Description of the scene, omitted here...
#### Driving Intensions: Description of the intension, omitted here...
#### Available actions: Description of the available actions, omitted here...
You can stop reasoning once you have a valid action to take.

**LLM response:**
I have 5 actions to choose from. Now, I would like to know which action is possible.
I should first check if I can accelerate, then idle, finally decelerate. I can also try to change lanes but with caution and not too frequently.
- I want to know if I can accelerate, so I need to observe the car in front of me on the current lane, which is car `952`. The distance between me and car `952` is 422.15 - 399.89 = 22.26 m, and the difference in speed is 14.08 - 24.03 = -9.95 m/s. Car `952` is traveling 22.26 m ahead of me and its speed is 9.95 m/s slower than mine. This distance is safe and my speed is already higher than the front car on the current lane. So I should not accelerate.
- Since I cannot accelerate, I want to know if I can idle. I need to observe the car behind me on the current lane, which is car `776`. The distance between me and car `776` is 399.89 - 411.68 = -11.79 m, and the difference in speed is 24.03 - 23.90 = 0.13 m/s. Car `776` is traveling 11.79 m behind me and its speed is 0.13 m/s slower than mine. This distance is too close and my speed is already higher than the rear car on the current lane. If I idle, the distance between me and car `776` will be further smaller, so I should not idle.
- In order to keep a safe distance with the rear car, I can only decelerate. Deceleration is a feasible action.
Great, I can make my decision now. Decision: Deceleration
Response to user:#### 4

**#2 Experience**

**Human Question:**
Here is the #3 scenario:
#### Driving scenario description: Description of the scene, omitted here...
#### Driving Intensions: Description of the intension, omitted here...
#### Available actions: Description of the available actions, omitted here...
You can stop reasoning once you have a valid action to take.

**LLM response:**
I have 5 actions to choose from. Now, I would like to know which action is possible.
I should first check if I can accelerate, then idle, finally decelerate. I can also try to change lanes but with caution and not too frequently.
- I want to know if I can accelerate, so I need to observe the car in front of me on the current lane, which is car `464`. The distance between me and car `464` is 532.46 - 464.36 = 68.10 m, and the difference in speed is 22.10 - 24.00 = -1.90 m/s. Car `464` is traveling 68.10 m ahead of me and its speed is 1.90 m/s slower than mine. This distance is safe and my speed is already higher than the front car on the current lane. So I should not accelerate.
- Since I cannot accelerate, I want to know if I can maintain my current speed. I need to observe the car in front of me on the current lane, which is car `464`. Due to above calculation, the distance between me and car `464` is 68.10 m, and the difference in speed is -1.90 m/s. The distance 68.10 m is higher than my preferred car following distance 30 m, and my speed is higher than the front car on the current lane. So I can maintain my current speed.
Great, I can make my decision now. Decision: Idle
Response to user:#### 1

**#3 Experience**

Figure 8: The 3-shot prompts example for the scenario shown in Figure 7. DiLu recall top three most similar experience from the memory module which contains 2 idle and 1 deceleration decision. The few-shot experience is entered into GPT-3.5 as a "human-LLM" dialogue.

```
System Prompts
You are ChatGPT, a large language model trained by OpenAI. Now you act as a mature driving assistant,
who can give accurate and correct advice for human driver in complex urban driving scenarios.
You will be given a detailed description of the driving scenario of current frame along with the
available actions allowed to take.

Make sure to include #### to separate every step.
```

```
Reflection Prompts
``` Human Message ```
{human_message}
``` ChatGPT Response ```
{llm_response}

Now, you know this action ChatGPT output cause a collison
after taking this action, which means there are some
mistake in ChatGPT resoning and cause the wrong action.

Please carefully check every reasoning in ChatGPT response
and find out the mistake in the reasoning process of
ChatGPT, and also output your corrected version of ChatGPT
response.

Your answer should use the following format:
#### Analysis of the mistake:
<Your analysis of the mistake in ChatGPT reasoning
process>
#### What should ChatGPT do to avoid such errors in the
future:
<Your answer>
#### Corrected version of ChatGPT response:
<Your corrected version of ChatGPT response>
```

```
Here is the current scenario:
#### Driving scenario description:
{scenario_description}
#### Driving Intensions:
{driving_intensions}
#### Available actions:
{available_actions}
You can stop reasoning once you have
a valid action to take.
```

```
I have 4 actions to choose from.
Now, I would like to know which
action is possible.
......
<reasoning process>
......

Response to user:#### ....
```

Figure 9: The prompts template for reflection module. The prompts in the grey box are fixed, while the prompts in the coloured box are different depending on the current scenario.

**Reflection prompts** Figure 9 presents the template for the reflection module's prompts, primarily comprising two sections: system prompts and reflection prompts. The system prompts section is entirely fixed and mainly consists of an introduction to the reflection task, along with instructions and formatting requirements for LLM responses. On the other hand, the reflection prompts section includes the scenario description of the erroneous decision and the faulty reasoning process made by the LLM. We require the reflection module to produce three components: an error analysis, corrected reasoning and decision-making, and suggestions on how to avoid making the same mistake in the future.

## A.3 CASE STUDY

In this section, let's demonstrate several case studies.

First, we present the results of the reasoning module for three cases, as shown in Figure 10. In Case 1, the green ego car is closely following the car 368 in front, and their speeds are similar. Initially, the driver agent explores whether it can accelerate in the current lane. The agent determines that, since the ego car's speed is similar to the car in front, there's no need to accelerate. Subsequently, the driver agent explores the possibility of maintaining the current speed. Through reasoning, the agent concludes that although the distance between the vehicles is slightly shorter than the ideal following distance, it's safe to maintain the speed because the ego car's speed is slightly lower than that of the car in front. Lastly, the agent verifies the feasibility of changing lanes to the right and calculates that the ego car's speed is lower than the car in the right lane by 5.88 m/s, with a safe following distance. Therefore, changing lanes to the right is also safe. In the end, the agent decides to change lanes to the right. In Case 2, the scenario is relatively simple. The agent observes a long distance with the car in front and that the ego car's speed is lower than that car. Consequently, it decides to accelerate. In this case, the agent does not perform further calculations regarding maintaining speed or changing lanes, aligning with the typical thought process and reasoning of human drivers. In Case 3, we changed the driving intention in the prompts from the default to "*I need to change to the rightmost lane.*" The driver agent recognizes this intent. Although it determines that it can maintain speed in the current lane, it continues to evaluate whether changing lanes to the right in the current lane is feasible. Ultimately, considering the intention, the agent chooses to change lanes to the right over maintaining speed.

**Case 1**

**Reasoning and Decision**

I have 5 actions to choose from. Now, I would like to know which action is possible.
I should first check if I can accelerate, then idle, finally decelerate. I can also try to change lanes but with caution and not too frequently.
- I want to know if I can accelerate, so I need to observe the car in front of me on the current lane, which is car `368`. The distance between me and car `368` is 558.08 - 528.53 = 29.55 m, and the difference in speed is 16.29 - 16.02 = 0.27 m/s. Car `368` is traveling 29.55 m ahead of me and its speed is 1.27 m/s faster than mine. This distance is safe and my speed is similar the front car on the current lane. So I should not accelerate.
- Since I cannot accelerate, I want to know if I can maintain my current speed. I need to observe the car in front of me on the current lane, which is car `368`. Due to above calculation, the distance between me and car `368` is 29.55 m, and the difference in speed is 0.27 m/s. The distance 29.55 m is larger than my preferred car following distance 30 m, but my speed is a bit lower than the front car on the current lane. So I can maintain my current speed.
- I can try to change lanes to the right, but I need to check if it is safe. I need to observe the car on the lane to my right, which is car `568`. The distance between me and car `568` is 541.40 - 528.53 = 12.87 m, and the difference in speed is 21.90 - 16.02 = 5.88 m/s. Car `568` is traveling 12.87 m ahead of me and its speed is 5.88 m/s faster than mine. This distance is safe and my speed is lower than the front car on the right lane. So I can change lanes to the right.
Great, I can make my decision now. Decision: **Turn-right**

**Case 2**

**Reasoning and Decision**

I have 5 actions to choose from. Now, I would like to know which action is possible.
I should first check if I can accelerate, then idle, finally decelerate. I can also try to change lanes but with caution and not too frequently.
- I want to know if I can accelerate, so I need to observe the car in front of me on the current lane, which is car `696`. The distance between me and car `696` is 817.67 - 773.04 = 44.63 m, and the difference in speed is 17.15 - 16.00 = 1.15 m/s. Car `696` is traveling 44.63 m ahead of me and its speed is 1.15 m/s faster than mine. This distance is safe and my speed is already lower than the front car on the current lane. So I can accelerate.
Great, I can make my decision now. Decision: **Acceleration**

**Case 3**

**Driving intention:** I need change to the rightmost lane to get off highway.

**Reasoning and Decision**

Based on the current scenario and the driving intentions, I will consider the available actions and refer to the previous successful examples to make a decision.
- I want to know if I can accelerate, so I need to observe the car in front of me on the current lane, which is car `424`. The distance between me and car `424` is 446.37 - 393.03 = 53.34 m, and the difference in speed is 21.28 - 25.21 = -3.93 m/s. Car `424` is traveling 53.34 m ahead of me and its speed is 3.93 m/s slower than mine. This distance is safe and my speed is already higher than the front car on the current lane. So I should not accelerate.
- Since I cannot accelerate, I want to know if I can idle. I need to observe the car in front of me on the current lane, which is car `424`. Due to the above calculation, the distance between me and car `424` is 53.34 m, and the difference in speed is -3.93 m/s. The distance 53.34 m is higher than my preferred car following distance 30 m, and my speed is higher than the front car on the current lane. So I can idle.
- Now, I need to consider my driving intention to change to the right lane. I should check the distance and speed of the cars on the right lane. The car on the right lane ahead of me is car `280`. The distance between me and car `280` is 429.23 - 393.03 = 36.20 m, and the difference in speed is 17.59 - 25.21 = -7.62 m/s. Car `280` is traveling 36.20 m ahead of me and its speed is 7.62 m/s slower than mine. This distance is safe and my speed is higher than the front car on the right lane. Therefore, it is safe to change to the right lane.
- Considering the above analysis and the driving intention, I can make my decision now.
Decision: **Turn-right**

Figure 10: Case study for reasoning module

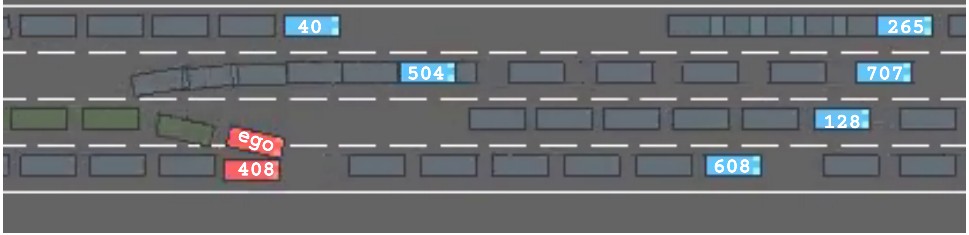

**Original Decision**
Based on the given scenario, let's analyze the situation and make the best driving decision.
- First, let's consider accelerating. The vehicle `128` is ahead of me in my current lane. It is at position 531.70 m, which is quite far from my current position of 480.91 m. The difference in speed between us is 18.19 - 19.34 = -1.15 m/s. Since the vehicle is far ahead and its speed is slower than mine, accelerating would only increase the speed difference and potentially lead to a collision. Therefore, accelerating is not a suitable option.
- Next, let's consider maintaining our current speed. The vehicle `128` is ahead of me in my current lane, and its speed is slightly slower than mine. If I maintain my current speed, I will gradually catch up to it, which may result in a collision. Therefore, maintaining my current speed is not a feasible option.
- Now, let's consider changing lanes. The vehicle on the right lane, `408`, is ahead of me. It is at position 481.25 m, which is relatively close to my current position of 480.91 m. The difference in speed between us is 17.82 - 19.34 = -1.52 m/s. The distance between us is appropriate, and its speed is slower than mine. Therefore, changing lanes to the right seems like a feasible plan to avoid a potential collision.
Based on the analysis, the best driving decision in this scenario is to change lanes to the right.
Decision: Turn-right

**Crash Analyze**
The mistake in ChatGPT's reasoning lies in its assessment of the feasibility of changing lanes to the right. It chose to change lanes to the right based on the slower speed of the vehicle (`408`) in the right lane and the distance between the two vehicles. However, it did not consider the relative position and speed of the vehicle in the right lane, which is crucial for safe lane changing. The vehicle in the right lane is only slightly ahead and moving slower, which means there's a high risk of collision if a lane change is attempted.

**Lesson Learned**
ChatGPT should consider not only the speed and distance of other vehicles but also their relative positioning when making decisions about lane changes. It's important to ensure that there's enough space and time to safely complete the lane change without causing a collision. This might involve more complex calculations of relative speed, distance, and time.

**Revised Decision**
Based on the given scenario, let's analyze the situation and make the best driving decision.
- First, let's consider accelerating. The vehicle `128` is ahead of me in my current lane. It is at position 531.70 m, which is quite far from my current position of 480.91 m. The difference in speed between us is 18.19 - 19.34 = -1.15 m/s. Since the vehicle is far ahead and its speed is slower than mine, accelerating would only increase the speed difference and potentially lead to a collision. Therefore, accelerating is not a suitable option.
- Next, let's consider maintaining our current speed. The vehicle `128` is ahead of me in my current lane, and its speed is slightly slower than mine. If I maintain my current speed, I will gradually catch up to it, which may result in a collision. Therefore, maintaining my current speed is not a feasible option.
- Now, let's consider changing lanes. The vehicle on the right lane, `408`, is ahead of me. It is at position 481.25 m, which is relatively close to my current position of 480.91 m. The difference in speed between us is 17.82 - 19.34 = -1.52 m/s. The distance between us is 481.25 - 480.91 = 0.34 m. Given the relative speed, the time to collision would be 0.34 / 1.52 = 0.22 seconds, which is too short to safely change lanes. Therefore, changing lanes to the right is not a feasible plan. Given these considerations, the safest course of action would be to decelerate, which would increase the distance between us and the vehicle ahead in the same lane, reducing the risk of collision.
Decision: Decelerate

Figure 11: Case study for reflection module

Next, in Figure 11, we present a result of the reflection module. In this case, the ego car is traveling in the second right lane and made the incorrect decision to change lanes to the right, resulting in a collision with car 408. Consequently, the reflection module intervenes to correct the mistake. First, it conducts an analysis of the cause of the collision. Upon reviewing the original reasoning process, GPT-4 astutely identifies the source of the error. It realizes that the initial decision did not take into account the relative distance between the vehicles in the right lane. In fact, car 408 was merely "slightly ahead" of the ego car, contrary to what the initial decision process described as an "appropriate" distance. Subsequently, in the revised decision process, the driver agent supplements and correctly calculates the relative position of the ego car and the car in the right lane. It also includes the calculation of "time to collision" (which was completely absent in the original decision). Based on this calculation, it determines that "time to collision is too short to safely change lanes." As a result, it chooses to decelerate, avoiding the collision. Finally, the reflection module summarizes the lesson learned from this error, emphasizing the importance of ensuring that there is enough space and time to safely complete a lane change without causing a collision: "*It's important to ensure that there's enough space and time to safely complete the lane change without causing a collision.*"

## A.4    Training settings of GRAD

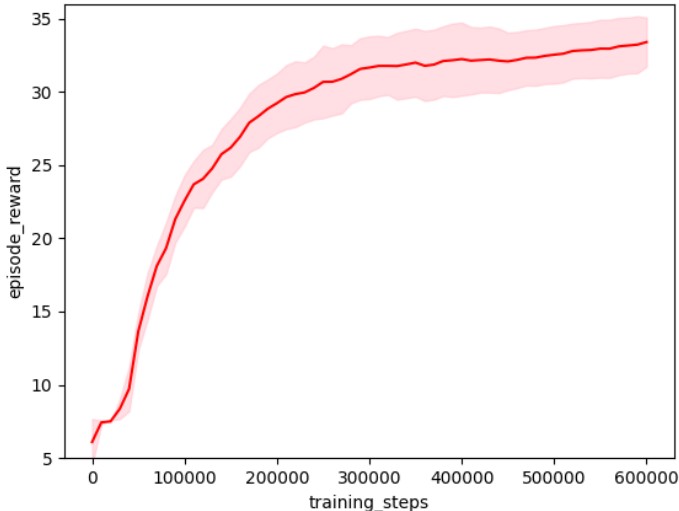

Figure 12: Rewards for each episode are evaluated during the training process. Each data point represents the average of 5 independent runs and has been smoothed using a window size of 9. The shadows correspond to the standard deviations of these runs.

Here we introduce the training settings of the RL method GRAD for the comparing with the DiLu. The rewards for each episode of GRAD are evaluated during the training process as shown in Figure 12. During the training, We set the Observations consisting of the kinematic status of the agent and the surrounding vehicles, specifically their coordinates, velocity, and heading. The maximum number of surrounding vehicles that can be observed is 32, and their speeds range from 15 to 25 units. As for the Action, it is discrete and comprises the following options: OneLaneLeft, Idle, OneLaneRight, SpeedLevelUp, and SpeedLevelDown. The available speed levels are 10, 15, 20, 25, and 32 units. The reward system includes two components, one is that a reward of 0.2 is granted for each time step survived by the agent, and the other one is that the reward is linearly mapped from the driving speed (10, 32), a reward value between 0 and 0.8. Moreover, Each model is trained for 600,000 action steps and is subsequently evaluated over 24 episodes using deterministic policies.

A.5    DIFFERENT DENSITY LEVEL

In order to give the reader an intuitive understanding of the different vehicle densities under Highway-env, we show in Figure 13 screenshots of a number of scenarios for the experimental part of this paper for three different traffic densities. It can be seen that with increasing vehicle density, there are more vehicles in the scenario, and the vehicle spacing and vehicle speed are reduced.

**4-lane 2.0 density**

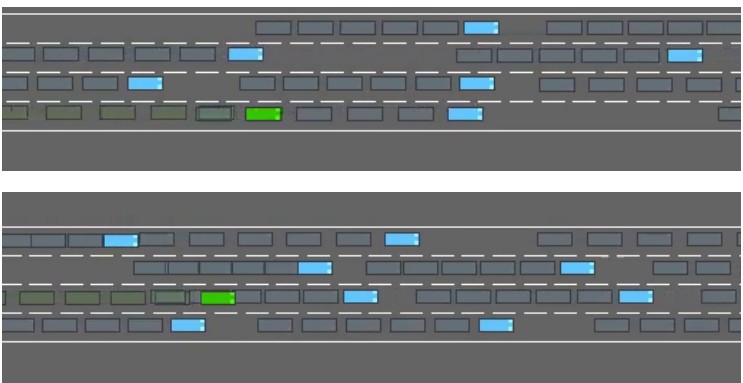

**5-lane 2.5 density**

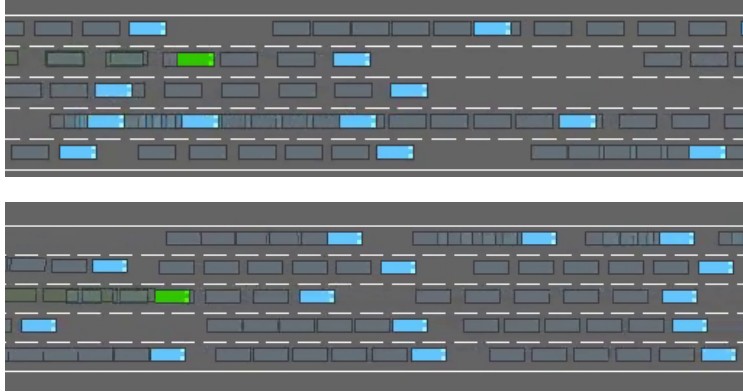

**5-lane 3.0 density**

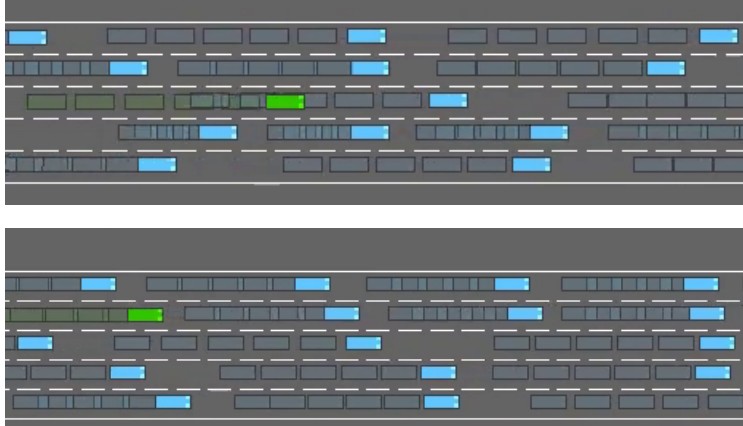

Figure 13: Different vehicle density in highway-env

