# OpenReview forum: "DiLu: A Knowledge-Driven Approach to Autonomous Driving with Large Language Models"
_ICLR.cc/2024/Conference — ICLR 2024 poster_

### Official Review · Reviewer_Cxkm · 2023-10-31

**Soundness:** 2 fair
**Presentation:** 2 fair
**Contribution:** 3 good
**Rating:** 6
**Confidence:** 4

**Summary:**

The paper includes a LLM into a framework which controls the decisions of a driving agent in a simulator. The authors define a concept, they call knowledge-based driving, and argue how their framework implements this and performs better than data-based methods. They test against one reinforcement learning based baseline in the Highway-env simulator.

**Strengths:**

The idea to use an LLM for scenario understanding and decision making in driving is very interesting. The authors have proposed a decent suggestion for integration and practically showed that it works.

The figures help getting a good top-level overview of the modules. Some parts are missing like how does the correction module work which is only an arrow in Figure 5?

**Weaknesses:**

The language could be clearer, less vague and heavily simplified to make the arguments easier to understand.

The evaluation against a single self-trained RL baseline makes it hard to estimate the performance. Since it seems there is no limit to the perception of the own agent, another fair comparison would be a simpler approach where a statistical or rule-based approach would have all information of all cars and drive. Without having a state-of-the-art RL method that is already optimized on highway-env it is hard to see if the performance gain comes from the proposed method or from the failure to adopt the RL method on the task.

Conceptually it is hard to imagine right now how this is supposed to drive in real time. Is the video sped up or slowed down? It is a challenge of the last decade how to get convolutional networks fast enough to be usable in a car. The computation challenges are not discussed at all. What is the reaction time of this and is execution speed a bottleneck?

**Questions:**

- What are the more precise concepts of knowledge-driven human driving that inspire this?
- Instill knowledge-driven capabilities sounds very vague. Methods that acquire experiences from real-world dataset covers all learning-based methods depending on what you mean with acquire. The abstract could be more concrete, it's hard to take away anything apart from that a LLM seems to do decision making while following a continuous learning scheme.

The language is hard to follow and the citations do not seem to support the claims well. In the Introduction, the sentence "This phenomenon inevitably leads to the marginal performance of data-driven methods." is one example for a broad claim without enough support in citations. There are autonomous cars driving in cities today with vision algorithms which are data-driven. They do not show marginal performance. The citations for this claim are one work describing a methodology to categorize corner cases in three common sensor modalities, so not very related, and the second citation "Chen et al. 2022" seems to be a catch all "survey of surveys" which is a large list of autonomous driving surveys with some added, partially trivial, thoughts.

Other examples where statements are too broad and hard to understand are: "Furthermore, this task is particularly formidable and expensive for autonomous driving systems due to the complex challenge of iterating diverse and unpredictable driving scenarios." What is this sentence supposed to say? The authors should heavily simplify their language to deliver their points clearer. Formidable and expensive can be understood in many different ways and distracts from the core argument the authors want to make.

Claims that the knowledge-based system is how a human drives can not be supported by the current state of research and not by the citations in this paper. I think the paper would benefit from not making the claim that they imitate a system in humans but limit themselves to saying, they designed a framework to include an LLM in a continuous learning setting where it outperforms certain other approaches.

Please add enough details from Johnson et al. 2019 to understand the vector similarity on an idea level. Make the paper more self-complete.

What is the data the LLM is trained on? If the training data contains driving situations from several countries, how to make sure it is following the appropriate traffic rules?

What are the 5 human crafted experiences and why are they needed?

Figure 7 a) could be easily replaced by a table to save space.

It is a bit unsatisfactory to have only a comparison against one baseline which was re-trained on this particular data. Is there no standard scenario on Highway-Env or another RL-based approach that was already applied to Highway-env to compare against? I could not find one myself so I don't see this as a downside in my rating but I think it would make the paper stronger if the authors could find a way to compare against more than one baseline.

---

> ### Author Response · Authors · 2023-11-17
> **Author Response for Reviewer Cxkm (Part 1)**
>
> Dear Reviewer Cxkm:
>
> Thanks for your time and constructive comments. We provide discussions and explanations about your concerns as follows.
>
> **Q1: The language could be clearer, less vague to make the arguments easier to understand.**
>
> **A1:** Thanks for your suggestion, we will polish the language in the paper in subsequent versions.
>
> **Q2: Without having a state-of-the-art RL method that is already optimized on highway-env it is hard to see if the performance gain comes from the proposed method or from the failure to adopt the RL method on the task.**  **Is there no standard scenario on Highway-Env or another RL-based approach that was already applied to Highway-env to compare against?**
>
> **A2:** We would like to clarify that our comparison method, GRAD (Graph Representation for Autonomous Driving) [1], is an RL-based approach that is **specifically optimized for autonomous driving scenarios and achieves SOTA performance on the highway-env environment.** GRAD creates a global scene representation that includes estimated future trajectories of other vehicles and outperforms the best social attention representation method in the highway-env environment. We will add more specific descriptions to GRAD within the experimental setup of the updated version.
>
> **Q3:  Is the video sped up or slowed down? The computation challenges are not discussed at all. What is the reaction time of this and is execution speed a bottleneck?**
>
> **A3:** The videos are not rendered in real-time. The decision-making process of the DiLu framework takes about 5-10 seconds, which includes the time for LLM inference and API call delays. As an initial attempt at a knowledge-driven AD framework, our goal in this paper is to demonstrate the capabilities of DiLu, such as causal reasoning, experience accumulation, and reflection for correction, without focusing too much on the performance aspects of the framework at this stage.
>
> In fact, there have been some advancements in both academia and industry regarding compressing LLMs and enhancing their inference speeds, such as reducing inference costs through linear transformations [2,3], and deploying smaller models on edge devices like GLM-2B[4], and Phi-1[5]. These explorations indicate the potential for further reductions in LLM execution times.
>
> **Q4: What are the more precise concepts of knowledge-driven human driving that inspire this?**
>
> **A4:**  The concept of "knowledge-driven" introduced in our paper contrasts with "data-driven". We view knowledge as a lower-dimensional, generalized representation of real-world phenomena, encompassing human knowledge and summaries of past experiences. Current data-driven AD systems learn the mapping between inputs (mostly sensor data, environmental context) and outputs (perception, decisions, controls), which lacks a comprehensive understanding of the environment and limits their generalization ability. In contrast, a Knowledge-driven AD system seeks to understand the surrounding environment and learn the causal reasoning capability between inputs and outputs through the common sense and knowledge. It also has the potential to be more interpretable, providing explanations for its decisions and actions.
>
> **Q5: Instill knowledge-driven capabilities sounds very vague.  The abstract could be more concrete, it's hard to take away anything apart from that a LLM seems to do decision making while following a continuous learning scheme.**
>
> **A5:** Thank you for your suggestions. Based on your valuable comments, we have revised the last part of the abstract as follows:
>
> > *To the best of our knowledge, we are the first to leverage knowledge-driven capability in decision-making for autonomous vehicles. Through the proposed DiLu framework, LLM is strengthened to apply knowledge and to reason causally in autonomous driving.*
>
> [1] Xi, Zerong, and Gita Sukthankar. "A Graph Representation for Autonomous Driving." *The 36th Conference on Neural Information Processing Systems Workshop*. 2022.
>
> [2] Katharopoulos, Angelos, et al. "Transformers are rnns: Fast autoregressive transformers with linear attention." *International conference on machine learning*. PMLR, 2020.
>
> [3] Peng, Bo, et al. "RWKV: Reinventing RNNs for the Transformer Era." *arXiv preprint arXiv:2305.13048* (2023).
>
> [4] Du, Zhengxiao, et al. "Glm: General language model pretraining with autoregressive blank infilling." *arXiv preprint arXiv:2103.10360* (2021). [https://huggingface.co/THUDM/glm-2b](https://huggingface.co/THUDM/glm-2b)
>
> [5] Gunasekar, Suriya, et al. "Textbooks Are All You Need." *arXiv preprint arXiv:2306.11644* (2023).

---

> ### Author Response · Authors · 2023-11-17
> **Author Response for Reviewer Cxkm (Part 2)**
>
> **Q6: In the Introduction, the sentence "This phenomenon inevitably leads to the marginal performance of data-driven methods." is one example for a broad claim without enough support in citations. There are autonomous cars driving in cities today with vision algorithms which are data-driven. They do not show marginal performance.**
>
> **A6:** To address your concerns, we accessed two common benchmarks used to evaluate vision algorithms for autonomous driving: [nuScenes detection](https://www.nuscenes.org/object-detection?externalData=no&mapData=no&modalities=Camera) and [SemanticKITTI segmentation](https://codalab.lisn.upsaclay.fr/competitions/6280#results). We have compiled statistics of SOTA methods and metrics from these benchmarks, starting from the first quarter of 2019 to the present, recorded quarterly. The results are shown in the table below. We also draw a line chart to provide a more intuitive view of the performance trends: [Performance Trend Chart (click it!)](https://smms.app/image/7rSgZh64kJLmcfe).
>
> |  | SemanticKITTI Lidar Segmentation |  | nuScenes 3D Detection |  |
> | --- | --- | --- | --- | --- |
> | Quarter |  Method | mIoU | Method | NDS |
> | 2019Q1 |  |  | PointPillars | 0.453 |
> | 2019Q2 | kechn | 0.525 | MEGVII | 0.633 |
> | 2019Q3 | flopie2009 | 0.537 | MEGVII | 0.633 |
> | 2019Q4 | hugues.thomas | 0.588 | MEGVII | 0.633 |
> | 2020Q1 | LiDARSeg | 0.613 | MEGVII | 0.633 |
> | 2020Q2 | Shuangjie | 0.637 | Noah CV Lab fusion | 0.689 |
> | 2020Q3 | MIT-HAN-LAB | 0.67 | Noah CV Lab lidar | 0.697 |
> | 2020Q4 | Cylinder3D | 0.689 | CenterPoint v2 | 0.713 |
> | 2021Q1 | AF2S3Net | 0.708 | CenterPoint v2 | 0.713 |
> | 2021Q4 | huanghui | 0.71 | Centerpoint-Fusion | 0.749 |
> | 2022Q1 | huanghui | 0.71 | FusionVPE | 0.754 |
> | 2022Q2 | Point-Voxel-KD | 0.734 | DeepInteraction-e | 0.762 |
> | 2022Q4 | UniSeg | 0.752 | MMFusion-e | 0.770 |
> | 2023Q1 | PointSeg | 0.753 | IEI-BEVFusion++ | 0.776 |
> | 2023Q2 | TASeg | 0.76 | IEI-BEVFusion++ | 0.776 |
> | 2023Q3 | TASeg | 0.76 | EA-LSS | 0.776 |
>
> It can be seen that the increase in the SOTA metrics for both benchmarks has slowed since 2021. In particular, nuScenes has improved by less than 3 percentage points from 2022 to the present. Furthermore, neither benchmark has crossed the 0.8 threshold in its respective metric. We believe that **data-driven algorithms, especially vision methods, do face a problem of marginal performance gains.**
>
> **Q7: "Furthermore, this task is particularly formidable and expensive for autonomous driving systems due to the complex challenge of iterating diverse and unpredictable driving scenarios." What is this sentence supposed to say?**
>
> **A7:** Sorry for the confusion, this sentence wants to convey that current autonomous driving systems require significant human labor and financial resources to collect and annotate driving data to handle complex and varied real-world driving scenarios. This process is essential for training and enhancing autonomous driving systems, but it presents a significant investment in terms of time, human labor, and costs.
>
> **Q8: I think the paper would benefit from not making the claim that they imitate a system in humans but limit themselves to saying, they designed a framework to include an LLM in a continuous learning setting where it outperforms certain other approaches.**
>
> **A8:** Thank you for your insight. We acknowledge your statement and will reduce the emphasis on imitating human behavior in the Introduction section of subsequent versions. Instead, we will focus more on designing the DiLu framework to enable the system to continuously accumulate experience and correct erroneous decisions made by LLMs, thereby continually improving performance.
>
> **Q9: Please add enough details from Johnson et al. 2019 to understand the vector similarity on an idea level.**
>
> **A9:** Thank you for your feedback! We’ve revised the content of Section 3.2, the Memory Module, to include an introduction to similarity retrieval.
>
> > *At each decision frame, the agent receives a textual description of the driving scenario. Before making a decision, the current driving scenario is embedded into a vector, which serves as the memory key. This key is then clustered and searched to find the closest scenarios (Johnson et al., 2019) in the memory module and their corresponding reasoning processes, or memories. These recalled memories are provided to the agent in a few-shot format to assist in making accurate reasoning and decisions for the current scenario.*
> >

---

> > ### Comment · Reviewer_Cxkm · 2023-11-20
> >
> > About Q6: The sentence in the paper speaks of "marginal performance of data-driven methods" which means absolute performance and not of marginal performance gains of segmentation and 3D detection in the last few years. The whole paragraph speaks of all data-driven methods in autonomous driving in order to contrast the author's concept of knowledge-driven driving which is a different and a much broader claim. It is not fair to data-driven methods or vision methods to pick a number like 0.8 of mIoU of one scene understanding dataset where this is concretely about Lidar segmentation and extrapolate even to the field of vision based algorithms. https://arxiv.org/pdf/2207.04397.pdf achieves 0.97 on cars btw. the detailed results are way more complex.
> >
> > This points to a general issue I see not resolved, the claims are too broad and don't even fit as motivation. If data driven vision methods are bad, how do you get your current frame to work on? It's as if you argue, your LLM is better than observing the road, it doesn't make sense if they need to work together in the end.
> >
> > The better choice to get a motivation would be the domain of scene understanding on a higher level, e.g. knowledge-graphs, planning, behavior prediction etc. Theoretically even Imitation Learning, specifically Behavioral Cloning could be a better field to compare. The LLM is reacting but is doing so because it should be able to react to close cars but understand the whole scenario. This understanding of a scenario is a hard problem and would be a much better motivator to contrast this work against.

---

> ### Author Response · Authors · 2023-11-17
> **Author Response for Reviewer Cxkm (Part 3)**
>
> **Q10: What is the data the LLM is trained on? If the training data contains driving situations from several countries, how to make sure it is following the appropriate traffic rules?**
>
> **A10:**  The LLM used in our DiLu framework includes GPT-3.5 and GPT-4 from OpenAI, which were trained on a vast corpus of internet text. Precisely due to the training data containing information about traffic scenarios and regulations in different countries, it is non-trivial to naively use LLM for decision-making in autonomous driving without any adaptations.
>
> DiLu framework addresses this issue through the use of a memory module and the few-shot approach. Specifically, the memory module stores experiences and knowledge applicable to the current country or scenario. When these related experiences are input into the large model in a few-shot manner, they also constrain the decision-making process to consider logic and rules relevant to the specific scenario.
>
> **Q11: What are the 5 human crafted experiences and why are they needed?**
>
> **A11:** We’re sorry for overlooking that detail in our article. We’ve updated the content of Section 3.2, the Memory Module, to include the specifics you mentioned.
>
> > *The initialization of memory module is similar to a human attending driving school before hitting the road. We select a few scenarios and manually outline the correct reasoning and decision-making processes for these situations to form the initial memory. These memories instruct the agent on the correct decision-making process for driving.*
> >
>
> **Q12: Figure 7 a) could be easily replaced by a table to save space.**
>
> **A12:** Thanks, we'll think about revising it.

---

> ### Author Response · Authors · 2023-11-21
> **Response to Reviewer Cxkm's Update**
>
> Thank you for your insightful feedback regarding our claims about data-driven methods in autonomous driving. We acknowledge that task-specific leaderboards are not sufficient to represent the absolute performance of data-driven methods in autonomous driving.
>
> In light of your feedback, we agree that **our initial statement about the "marginal performance of data-driven methods" was overly broad**. It did not effectively distinguish between the absolute performance of these methods and the incremental performance gains observed in recent years.
>
> To address this, we **have removed the original sentence,** "This phenomenon inevitably leads to the marginal performance of data-driven methods" and revised the first paragraph of the Introduction as follows:
>
> > *These data-driven algorithms strive to capture and model the underlying distributions of the accumulated data (Bolte et al., 2019; Zhou & Beyerer, 2023), encountering challenges such as dataset bias, overfitting, and uninterpretability (Codevilla et al., 2019; Jin et al., 2023). **Exploring methods to mitigate these challenges could lead to a deeper understanding of driving scenarios and more rational decision-making, potentially enhancing the performance of autonomous driving systems.***
> >
> This revision aims to better clarify our research motivation and align with the context of exploring higher-level scene understanding to enhance decision-making, as you suggested.
>
> Thank you again for your valuable suggestions on how to refine our manuscript and make it more precise.

---

### Official Review · Reviewer_48LR · 2023-11-01

**Soundness:** 3 good
**Presentation:** 3 good
**Contribution:** 3 good
**Rating:** 6
**Confidence:** 4

**Summary:**

The paper presents a framework for utilizing the few shot and self-correction capabilities of LLMs for the task of AV planning, where the following abilities of the framework are highlighted:
- Store successful experiences in memory and leverage them to improve future rollouts through similarity retrieval and usage in few-shot prompting.
- Ability to learn from unsuccessful experiences (ones with collisions) by applying LLM self-correction and storing the modified experience among the successful experiences in the memory

The above components, dubbed as reasoning and reflection modules respectively, are integrated along with memory in a closed loop setting without any back-propagation objective.

A number of prompting techniques including chain-of-thought and few-shot prompting are used to get better reasoning. The environment used for experiments (Highway-Env) only requires four discrete decisions, hence the LLM is prompted to select one amongst these four decisions for each frame after going through CoT reasoning.

The experiments are used to demonstrate the following key claims:
- The memory module combined with few-shot prompting provides much better results than using no memory module (zero-shot) or using lesser shots.
- The more the number of experiences in the memory, the better.
- The ability to generalize is better with more few-shot experiences fed into the LLM
- Adding successful and corrected experiences both help in improving performance
- Better generalization capability compared to RL method GRAD.

**Strengths:**

- The motivating idea of human knowledge distillation for AV planning is sound, interesting, and under-explored.

- The overall framework formulation towards leveraging LLMs via appropriate prompting, retrieval, and self-correction is interesting and well set up. It would have been exciting to see formulations for LLMs assisting planning stacks (instead of directly doing discrete action decision making) - which could be much more valuable to existing systems.

- The flywheel effect created from storing both successful and unsuccessful + corrected experiences in memory is an important contribution.

- The paper provides a good foundation for other exciting work to build upon, especially with the promise to open source upon acceptance.

- The experiments are fairly extensive towards investigating all the different components of the proposed framework.

**Weaknesses:**

-  One of the main proposed advantages is better generalization through instilling human knowledge-driven capabilities instead of a data-driven only approach. However, the experimental settings derived from HighwayEnv are too restrictive to help extrapolate how such LLM based reasoning would perform on diverse new scenes using retrieval + few shot prompting. While it is perfectly fine to work with restricted settings and smaller datasets for new research work, the bridge to answer the most interesting questions is too long.

- As mentioned in strengths section, providing directions and initial experiments on assisting planning stacks (instead of directly doing discrete action decision making) could provide a lot of value.

- The experiment settings used to demonstrate generalization are not too convincing. The number of lanes and traffic density is changed, but this is still an extremely similar environment where the retrieved few-shot scenarios could be nearly directly applicable.

- Under the above setting, it is possible that with a large enough memory module the task reduces to simply copying the answer from one of the retrieved experiences. It would be good to see a baseline where the decision from one of the retrieved experiences is used as is (voting with mixture of experts or winner takes all)

- The metric movement with CitySim in Figure 7b and Table 1 correction row do not seem significant to make the corresponding claims?

- Nit: The key frame sampling for successful experiences seems like an important detail that has not been explained.

- Minor nit: The claim for this being the first work addressing AV planning via leveraging LLMs might need to be revised with recent papers like GPT driver (depending on chronology).

**Questions:**

- What kind of diverse interactions do we get from the Highway-env simulator? Would it be possible to evaluate the framework under more interactive / challenging conditions, especially wrt agent interactions? It would be interesting to see the generalization to intersections, interactions with peds, aggressive agents etc.

- The correction experiences intuitively should provide a strong boost to performance since they are akin to hard example mining and injecting reasoning about the negative outcomes. However the corresponding results in Table 1 do not show strong improvements. Is it possible understudied and warrants more extensive experimentations?

---

> ### Author Response · Authors · 2023-11-17
> **Author Response for Reviewer 48LR (Part 1)**
>
> Dear Reviewer 48LR:
>
> Thanks for your time and insightful suggestions. We provide discussions and explanations about your concerns as follows.
>
> **Q1:  The experimental settings derived from HighwayEnv are too restrictive to help extrapolate how such LLM based reasoning would perform on diverse new scenes. The bridge to answer the most interesting questions is too long.**
>
> **A1:**  Although the experimental setup in highway-env has certain limitations, we have demonstrated the effectiveness of the DiLu framework through our experiments, including the accumulation of memory and the use of few-shot experiences, as well as its outstanding generalization and transformation capabilities compared to RL-based methods. We also validated the importance of the two types of memories in the Reflection module through ablation experiments. Furthermore, we supplemented our study with experiments on aggressive agents, as detailed in the response to Q8.
>
> We are excited about applying the knowledge-driven paradigm to autonomous driving, as this is our initial work. We will continue to explore how knowledge can be used to drive the entire autonomous driving pipeline (including perception, decision making, and planning) so that this paradigm can be applied in a more diverse and broader range of scenarios.
>
> **Q2: Providing directions and initial experiments on assisting planning stacks could provide a lot of value.**
>
> **A2:** Your suggestion is insightful. Using LLMs for decision making and further trajectory planning in continuous space is promising. In fact, we have tried to enable DiLu to perform trajectory planning, but the performance has not been satisfactory. One reason for this is that LLMs have difficulty producing detailed temporal trajectories, which are a series of coordinate points. Also, the trajectories produced by LLMs may contain certain hallucinations, making these output trajectories potentially risky. We believe that using existing planning methods (such as sample-based or search-based methods) after the decision process is complete seems more feasible. This will be one of the directions we will explore in our future work.
>
> **Q3: The experiment settings used to demonstrate generalization are not too convincing. The number of lanes and traffic density is changed, but this is still an extremely similar environment.**
>
> **A3:** The increase in the number of lanes and traffic density actually leads to changes in driving strategies. For example, increased traffic density tends to slow down the average speed of the traffic flow, requiring more cautious driving, while an increased number of lanes provides the ego vehicle with more lane-changing opportunities. As shown in Figure 7(a), these changes in the experimental settings have a significant impact on the performance of the RL-based method. In addition, to further validate the generalization capabilities of DiLu, we perform additional experiments in environments with aggressive agents (as detailed in the response to Q8).
>
> **Q4:  It would be good to see a baseline where the decision from one of the retrieved experiences is used as is (voting with mixture of experts or winner takes all)**
>
> **A4:**  To address your concern To address your concern, we conducted additional tests using a winner-take-all baseline method with the following setup: We used a memory module containing 160 experiences (including those accumulated from the Highway-env and those obtained from the CitySim dataset). For each decision frame, we conduct a similarity query for the five most similar experiences from the memory module. If the decision outcomes of at least *three of the five* experiences were consistent, we used that outcome for the decision of the current frame. If fewer than three experiences had the same decision outcome, we still used a few-shot approach for the LLM-based decision. We conducted ten repeated experiments, and here are the results:
>
> | Framework | Success rate (SR) | Avg. Success steps (SS) | Median Success steps (SS) |
> | -- | -- | -- | -- |
> | Winner takes all, with 160 memories | 60% | 22.5 | 30 |
> | DiLu, with 40 memories | **90%** | **27.7** | **30** |
>
> Despite having a knowledge base four times larger than that of DiLu, the winner-takes-all baseline *underperformed* compared to the DiLu method in terms of SR (Success Rate) and Avg. SS (Average Sequential Success). This is because the similar experiences retrieved from the memory module cannot be simply copied as answers. For instance, two very similar scenarios might require entirely different decisions due to minor differences in the acceleration of the car ahead, and directly copying answers could lead to dangerous situations.
>
> In contrast, the few-shot approach used by the DiLu framework leverages the common sense and causal reasoning capabilities inherent in LLMs. It learns the key knowledge and reasoning logic from few-shot experiences, thereby making more reasonable decisions.

---

> ### Author Response · Authors · 2023-11-17
> **Author Response for Reviewer 48LR (Part 2)**
>
> **Q5: The metric movement with CitySim in Figure 7b do not seem significant to make the corresponding claims?**
>
> **A5:** Figure 7 (b) exhibits the transformation performance of DiLu using knowledge from real-world dataset (CitySim). It can be found that DiLu performs better with knowledge from CitySim dataset compared with one without any prior experiences (the righty-side boxes are higher than the gray dash-dot line). Also, when tested in a more congested traffic environment, the knowledge in CitySim obtains better performance (Max SS=30, right fleshcolor box) compared with the accumulated experience in the simulation domain (Max SS=23, left fleshcolor box).
>
> **Q6: Nit - The key frame sampling for successful experiences seems like an important detail that has not been explained.**
>
> **A6:** We indeed overlooked the details of keyframe sampling in our text. In our experiments, we uniformly sample frames in the sequence at equal intervals. However, this is merely a simple setup for our experiments. A more effective approach would be to cluster the scene descriptions of these frames and then select a frame from each cluster for storage.
>
> **Q7: NIT - The claim for this being the first work addressing AV planning via leveraging LLMs might need to be revised with recent papers like GPT driver (depending on chronology).**
>
> **A7:** We are also aware of recent works using LLMs for solving autonomous driving planning tasks, such as GPT-Driver[1], DriveGPT4[2], LanguageMPC[3], and Driving with LLMs[4]. However, all these works were published after October 1st, 2023, which is after the ICLR submission deadline (September 28th, 2023). To ensure accuracy and precision, we have revised the relevant statements in the Abstract and Introduction to uniformly state: "We are the first to leverage knowledge-driven capability in decision-making for autonomous vehicles."
>
> **Q8: What kind of diverse interactions do we get from the Highway-env simulator? Would it be possible to evaluate the framework under more interactive / challenging conditions, especially wrt agent interactions?**
>
> **A8:** We agree with your point that interactions between agents present a more challenging experimental setup. In response to this, we have performed new experiments with different types of agents. The default behavior model of surrounding cars in Highway-env is [IDMVehicle](https://highway-env.farama.org/dynamics/vehicle/behavior/#highway_env.vehicle.behavior.IDMVehicle), and we added experiments using [AggressiveVehicle](https://highway-env.farama.org/dynamics/vehicle/behavior/#highway_env.vehicle.behavior.AggressiveVehicle) and [DefensiveVehicle](https://highway-env.farama.org/dynamics/vehicle/behavior/#highway_env.vehicle.behavior.DefensiveVehicle) behavior models. These vehicle types exhibit more aggressive lane changing and acceleration/deceleration behaviors, thus posing greater challenges to the ego car's decision-making. We conducted experiments with both DiLu and the RL-based approach (GRAD) in environments with these three types of vehicle behaviors, using a 5-shots setting, `lane-4-density-2`, and 40 experiences in the memory module.
>
> | Environment | Success Rate (SR) | Avg. Success steps (SS) | Median Success steps (SS) |
> | --- | --- | --- | --- |
> | IDM Vehicle, DiLu | **90%** | **29.2** | **30** |
> | IDM Vehicle, GRAD | 70% | 26.1 | 29 |
> | Aggressive Vehicle, DiLu | **80%** | **26.2**| **30** |
> | Aggressive Vehicle, GRAD | 48% | 20.1 | 23.1 |
> | Defensive Vehicle, DiLu | **70%** | **22.5** | **30** |
> | Defensive Vehicle, GRAD | 40% | 19.7 | 21.5 |
>
> The results show that changes in the behavior type of surrounding vehicles significantly decrease the success rate of the RL-based method (70%->40%), while DiLu's performance, although slightly reduced, is still able to successfully drive in over 70% of cases for a continuous 30 steps. This demonstrates the clear advantage of our proposed knowledge-driven framework under more interactive and challenging conditions. **We have uploaded the new experiment videos in the supplementary materials.**
>
> Due to the limited rebuttal period and the limitations of Highway-env, we were unable to complete experiments involving interactions at intersections or with pedestrians. As one of our next steps, we are exploring alternatives to Highway-env and conducting experiments in more realistic and diverse simulators.
>
>
> [1] Mao, Jiageng, et al. "Gpt-driver: Learning to drive with gpt." *arXiv preprint arXiv:2310.01415* (2023).
>
> [2] Xu, Zhenhua, et al. "DriveGPT4: Interpretable End-to-end Autonomous Driving via Large Language Model." *arXiv preprint arXiv:2310.01412* (2023).
>
> [3] Sha, Hao, et al. "Languagempc: Large language models as decision makers for autonomous driving." *arXiv preprint arXiv:2310.03026* (2023).
>
> [4] Chen, Long, et al. "Driving with llms: Fusing object-level vector modality for explainable autonomous driving." *arXiv preprint arXiv:2310.01957* (2023).

---

> ### Author Response · Authors · 2023-11-17
> **Author Response for Reviewer 48LR (Part 3)**
>
> **Q9: The corresponding results in Table 1 do not show strong improvements. Is it possible understudied and warrants more extensive experimentations?**
>
> **A9:** We need to clarify that the last row of Table 1 presents the results of using **both success memory and correction memory simultaneously**. We provide the following table to remove any ambiguity:
>
> | Methods | MIN | Q1 | Median | Q3 | MAX |
> | --- | --- | --- | --- | --- | --- |
> | baseline | 2.0 | 3.0 | 10.0 | 22.0 | 30.0 |
> | baseline + success memory | 3.0 | 7.0 | 24.5 | 29.3 | 30.0 |
> | baseline + correcction memory | 4 | 7.75 | 21.5 | 30 | 30 |
> | **baseline + success & correcction memory** | **5** | **7.75** | **24.5** | **30** | **30** |
>
> The purpose of the ablation study and Table 1 is to show that both types of memory are useful. However, it also shows that there is some overlap in the knowledge contained in the success and correction memories. Therefore, the improvement brought about by the correction memory is not as significant as the reviewer might expect.

---

### Official Review · Reviewer_RmuT · 2023-11-06

**Soundness:** 3 good
**Presentation:** 3 good
**Contribution:** 3 good
**Rating:** 8
**Confidence:** 3

**Summary:**

This work proposes a novel and interesting approach leveraging LLMs in autonomous driving to perform knowledge-based reasoning about making high level driving decisions. The approach is motivated by how humans learn to drive. There are three straightforward pieces of the method: reasoning, recall, and reflection. The method is evaluated in simulating driving scenarios and positively compared against a SOTA RL method and ablations of the approach.

**Strengths:**

The proposed method is well-motivated by human behavior and generally clearly explained. The experiments justify each portion of the method for achieving the goal task of autonomous driving.  The method is novel, simple, and has potential to be used in the real world. Overall, an interesting perspective on the self-driving car problem.

**Weaknesses:**

The memory module requires more description in Section 3.2. The process of storing experiences is somewhat unclear. The writing could be interpreted to mean that every scenario is stored separately or that the similarity between the keys is used to map similar experiences to the same memory store (which seems to be what the authors are actually doing). Either a new figure or updates to the existing figures would also add to clarity and precision.

This paper never discusses limitations. I strongly recommend making room to discuss the relationship between this approach and approaches which focus on safety. In fact, the “reflection” module is being presented as a safety mechanism. However, the trustworthiness of the results from the LLM is never discussed. Diving into reliability and limitations is important in a method which claims to address safety for transparency in a safety critical task where results are currently deployed in the real world.

I thought the following claim in the abstract was slightly misleading given LINGO-1 (which the authors do cite). “To the best of our knowledge, we are the first to instill knowledge-driven capability into autonomous driving systems from the perspective of how humans drive.” I think that the correct way to phrase what the authors mean is specifically saying that they are the first to “use human-like knowledge-based reasoning to make autonomous driving decisions” or something similar since leveraging it in decision making is the distinction with prior work. “Instill” is a vague term which could also describe what LINGO-1 is doing.

**Questions:**

It is fairly odd in the experiments that two different GPT versions are used. Why did the authors not just use GPT-4?

---

> ### Author Response · Authors · 2023-11-17
> **Author Response for Reviewer RmuT**
>
> Dear Reviewer Rmut:
>
> Thank you for your constructive comments. We provide discussions and explanations about your concerns as follows.
>
> **Q1: The memory module requires more description in Section 3.2. The process of storing experiences is somewhat unclear.**
>
> **A1:** Thank you for your suggestion. As you’ve mentioned, the DiLu framework stores memories individually for each scene. It’s during the recall process that it searches for similar scenes and retrieves the corresponding experiences. We will revise the article to more clearly detail this process in Section 3.2. Part of the revised text is as follows:
>
> > *The memory stored in the memory module consists of two parts: scene descriptions and corresponding reasoning processes.
> As the agent makes correct reasoning and decisions, or reflects on the correct reasoning process, it gains driving experience. We embed the scene description into a key, pair it with the reasoning process to form a memory and store it in the memory module.*
> >
>
> **Q2: This paper never discusses limitations. In fact, the “reflection” module is being presented as a safety mechanism. However, the trustworthiness of the results from the LLM is never discussed.**
>
> **A2:** We appreciate your insightful suggestions, and will incorporate a section on model limitations in the paper.
>
> > *The issue of LLM hallucinations and how to deal with them has indeed been widely recognized and researched [1,2]. Our experiments have shown that relying solely on driving knowledge for LLMs to make zero-shot driving decisions is highly ineffective. DiLu provides LLMs with driving knowledge through few-shot learning. This significantly reduces the occurrence of hallucinations and improves DiLu's driving performance. However, it's important to note that these methods can only improve safety, not eliminate errors completely. In the DiLu framework, we employ LLM as an agent due to its ability to extract and utilize knowledge from text. This enables us to experimentally demonstrate the feasibility of knowledge-driven autonomous driving. In the future, as LLMs become more advanced and illusions are progressively reduced, or as more stable models capable of extracting knowledge from text emerge, they could also serve as agents in the DiLu framework, potentially improving performance.*
> >
>
> **Q3: I thought the following claim in the abstract was slightly misleading given LINGO-1 (which the authors do cite). “To the best of our knowledge, we are the first to instill knowledge-driven capability into autonomous driving systems from the perspective of how humans drive.”. “Instill” is a vague term which could also describe what LINGO-1 is doing.**
>
> **A3:**  We agree with your viewpoint. The phrase "instill knowledge-driven capability into autonomous driving" is indeed somewhat vague. We have decided to revise it to "We are the first to **leverage knowledge-driven capability in decision-making for autonomous vehicles**," to ensure that our statement is sufficiently precise.
>
> **Q4: It is fairly odd in the experiments that two different GPT versions are used. Why did the authors not just use GPT-4?**
>
> **A4:** The main reason for using GPT-4 only in the Reflection module and GPT-3.5 in the Reasoning module was the lower cost and faster response time of the GPT-3.5 API (about 3 to 6 times faster than GPT-4, as mentioned [in this link](https://community.openai.com/t/gpt-3-5-and-gpt-4-api-response-time-measurements-fyi/237394)). Certainly, the DiLu framework fully supports just using GPT-4 as the LLM. In response to your question, we conducted additional tests using the latest version of GPT-4 in all modules of the DiLu framework and **achieved better results than those reported in the original paper**. The results are shown in the table below:
>
> | Success steps (SS) | Average SS | Min SS |
> | --- | --- | --- |
> | GPT-3.5 in reasoning module (original version) | 27.7 | 7 |
> | GPT-4 in reasoning module (new tested) | **29.2** | **26** |
>
> *: Tested in `lane-4-density-2` setting with 5-shots.
>
> [1] Wei, Jason, et al. "Chain-of-thought prompting elicits reasoning in large language models." *Advances in Neural Information Processing Systems* 35 (2022): 24824-24837.
>
> [2] Yao, Shunyu, et al. "Tree of thoughts: Deliberate problem solving with large language models." arXiv preprint arXiv:2305.10601 (2023).

---

> > ### Comment · Reviewer_RmuT · 2023-11-21
> > **Reply to rebuttal**
> >
> > Thank you for the additional clarifications here and to the other reviewer comments. After reading through, I have increased my score.

---

> ### Author Response · Authors · 2023-11-22
>
> Thanks a lot for your acknowledgement, and we appreciate the time and effort you dedicated to enhancing the quality and clarity of our manuscript.
>
> Your support and recognition is greatly valued.

---

### Official Review · Reviewer_LJSU · 2023-11-10

**Soundness:** 3 good
**Presentation:** 3 good
**Contribution:** 2 fair
**Rating:** 5
**Confidence:** 4

**Summary:**

This paper presents a novel framework for autonomous driving systems based on LLM and tailored components. Contributions of this paper are several folds:

- Knowledge-Driven Paradigm: The paper introduces a knowledge-driven paradigm for autonomous driving, differentiating it from existing data-driven approaches. This paradigm is inspired by human driving, which relies more on knowledge and understanding rather than mere data accumulation.

- DiLu Framework: The authors propose the DiLu framework, integrating large language models (LLMs) with autonomous driving systems. Several modules are proposed based on recent advances of AI agent: A Reasoning Module that utilizes LLMs for decision-making based on common-sense knowledge; A Reflection Module that assesses decisions and updates them based on safety and correctness, using the knowledge from LLMs.

- Experimentation and Results: Extensive experiments demonstrate the framework's capability to make proper decisions, its strong generalization ability, and the potential for real-world application. The paper compares DiLu with reinforcement learning methods, showing its superior performance in generalization and adaptability.

**Strengths:**

- Innovative Approach: The integration of LLMs into autonomous driving systems represents a significant shift from traditional data-driven methods, potentially offering more adaptable and human-like decision-making.

- Generalization Ability: DiLu shows a strong ability to generalize from one environment to another, a crucial aspect for real-world applicability.

- Continuous Learning: The framework's ability to continuously evolve and improve through its memory and reflection modules is a key strength.

**Weaknesses:**

- Complexity and Scalability: The integration of LLMs and the need for continuous updating and reflection may introduce complexity, potentially impacting the scalability of the system.

- Real-World Application: While the framework shows promise, the transition from controlled experiments to real-world application can be challenging, given the unpredictable nature of real-world environments.

- Dependence on LLMs: The framework's reliance on LLMs means that its performance is heavily dependent on the capabilities and limitations of these models.

- Evaluation thoroughness: The authors only evaluate the proposed methods with oversimplied metrics (collisions) and compared to a simple baseline (RL). The limitation of the evaluation poses a question mark on how such system actually performs in the real driving scenarios, compared to sota autonomous driving systems.

**Questions:**

While LLM-based agent systems have shown success in various embodied systems, the adaptation of it in the AV tasks is still unclear to the reviewer. AI agent system has shown prominent success in task planning for open world robotic tasks, but AV has a different setting (with different challenges). The motivation and advantages of using AI agent system for AV needs to be elaborate more.
On the other hand, the authors didn't evaluate the proposed framework thoroughly enough (with only simple metrics and simple baselines). This further raises questions of the reviewer regarding how promising or what are the key advantages of using AI agent system in AV setting.
Finally, the proposed AI agent follows a typical setup compared to the other existing works in robotics tasks. The authors should highlight more on the unique challenges and design choices tailored for the AV task.

---

> ### Author Response · Authors · 2023-11-17
> **Author Response for Reviewer LJSU  (Part 1)**
>
> Dear Reviewer LJSU：
>
> Thank you for your comments. We provide discussions and explanations about your concerns as follows.
>
> **Q1: Complexity and Scalability - The integration of LLMs and the need for continuous updating and reflection may introduce complexity, potentially impacting the scalability of the system.**
>
> **A1:** We understand your concerns about introducing LLMs potentially increasing system complexity and presenting scalability issues. In fact, the DiLu framework is designed to address these problems in several ways: we store driving experiences in the memory module, and each time only a small number of similar experiences are looked up for few-shot learning, limiting the dispersion of LLM responses and enhancing the ability of LLMs to solve AD tasks. The memory module can simultaneously contain driving experiences from different datasets for different traffic scenarios. Moreover, different LLMs can be integrated into the DiLu framework. The textually stored driving experiences/knowledge can be immediately input to new LLMs in the proposed few-shot approach, without the need to accumulate from scratch again.
>
> **Q2: Real-World Application - While the framework shows promise, the transition from controlled experiments to real-world application can be challenging.**
>
> **A2:** Our paper, as an early work on leveraging human knowledge reasoning capabilities in autonomous driving, focuses more on validating the capabilities of the proposed DiLu framework, including using LLMs for driving decisions and accumulating driving experience through the memory module. How to actually apply large language models in autonomous vehicles, such as the reduction of inference costs through linear transformations [1,2] and the deployment of small models on edge devices (e.g. GLM-2B[3], DeciCoder[4], Phi-1[5]), is an ongoing exploration in the industry and is beyond the scope of this paper. Furthermore, given the unpredictable nature of the real world, it is precisely the common sense knowledge contained in LLMs that the driver agent needs to make decisions, rather than data-driven  methods.
>
> **Q3: Dependence on LLMs - The framework's reliance on LLMs means that its performance is heavily dependent on the capabilities and limitations of these models.**
>
> **A3:** This is precisely the advantage of our proposed DiLu framework over those methods that require fine-tuning of LLMs. DiLu uses out-of-the-box LLMs and is loosely coupled to them. As large models continuously evolve (e.g. from GPT-3.5 to GPT-4 to GPT-4 Vision), our framework can take advantage of the improved capabilities of these models without any modifications. At the same time, the driving experience previously accumulated in the memory module (represented in pure text) can be fed into new LLMs, eliminating the need to start accumulating from scratch.
>
> **Q4:  While LLM-based agent systems have shown success in various embodied systems, the adaptation of it in the AV tasks is still unclear to the reviewer. The motivation and advantages of using AI agent system for AV needs to be elaborated more.**
>
> **A4:** One of the motivations of our paper is that the preliminary results achieved by LLM-based agents in the field of embodied AI have shown us the potential for their application in AV tasks. As an initial attempt to apply LLMs in the field of autonomous driving, our paper proposes the novel DiLu framework. DiLu continuously accumulates knowledge in the driving domain through its memory module and improves the closed-loop decision performance with few-shot driving experiences. In addition, DiLu uses a reflection module to reflect and correct erroneous decisions, allowing the whole framework to continuously improve its performance by exploring the environment.
>
>
> [1] Katharopoulos, Angelos, et al. "Transformers are rnns: Fast autoregressive transformers with linear attention." *International conference on machine learning*. PMLR, 2020.
>
> [2] Peng, Bo, et al. "RWKV: Reinventing RNNs for the Transformer Era." *arXiv preprint arXiv:2305.13048* (2023).
>
> [3] Du, Zhengxiao, et al. "Glm: General language model pretraining with autoregressive blank infilling." *arXiv preprint arXiv:2103.10360* (2021). [https://huggingface.co/THUDM/glm-2b](https://huggingface.co/THUDM/glm-2b)
>
> [4] DeciCoder, DeciAI Research Team, [https://huggingface.co/Deci/DeciCoder-1b](https://huggingface.co/Deci/DeciCoder-1b)
>
> [5] Gunasekar, Suriya, et al. "Textbooks Are All You Need." *arXiv preprint arXiv:2306.11644* (2023).

---

> ### Author Response · Authors · 2023-11-17
> **Author Response for Reviewer LJSU (Part 2)**
>
> **Q5: The authors didn't evaluate the proposed framework thoroughly enough, with oversimplified metrics (collisions) and compared them to a simple baseline (RL). This further raises questions of the reviewer regarding how promising or what are the key advantages of using AI agent system in AV setting.**
>
> **A5:** We understand the reviewer's concern about the thoroughness of the framework evaluation. We would like to clarify the following points:
>
> 1. The baseline algorithms we compared are not just simple reinforcement learning methods. GRAD, which stands for [Graph Representation for Autonomous Driving,](https://ml4ad.github.io/files/papers2022/A%20Graph%20Representation%20for%20Autonomous%20Driving.pdf) is an RL-based approach specifically designed for autonomous driving scenarios [6]. GRAD generates a global scene representation that includes estimated future trajectories of other vehicles. It outperforms the best-performing social attention representation methods in the highway environment.
> 2. Evaluation metrics in the experimental section include Success Rate (SR) and Sequential Success Steps (SS). We define "*a successfully completed task means that the ego vehicle navigates through traffic at a reasonable speed and without collisions*" (in Appendix A.1), which includes two metrics commonly used in closed-loop simulations[7]: No collision and Speed limit compliance.
>
> Through this comparative experiment, we aim to demonstrate the core advantage of the DiLu framework: its use of common sense knowledge and minimal experience accumulation to make reasonable decisions in autonomous driving. As mentioned in Section 4.3, the DiLu framework needs only 40 memories to demonstrate comparable performance to a reinforcement learning method trained on 600,000 episodes in training scenarios, and much higher generalization capability compared to data-driven methods.
>
> **Q6: The authors should highlight more on the unique challenges and design choices tailored for the AV task.**
>
> **A6:** Robotics tasks tend to have clearer objectives (such as grasping, navigation, etc.) and researchers are more concerned with completing the entire task. Most of the scenarios that robots are exposed to are closed, low-speed indoor scenarios. In contrast, the objectives of autonomous driving tasks are more ambiguous (just driving on the road) and require consideration of more complex performance metrics (including driving safety, passenger comfort, and compliance with traffic regulations).
>
> The unique challenges of Autonomous Vehicle (AV) tasks are mainly related to the fact that AVs operate in highly dynamic and open environments and encounter more unexpected edge cases. These challenges are difficult to address with traditional rule- or RL-based methods. The proposed DiLu, through its memory module, records past valuable experiences and leverages the common sense knowledge inherent in LLMs (i.e., traffic rules, how to properly interact with other participants) to achieve better generalization and continuous learning. We thank you for the insightful suggestion and will consider adding these explanations to the introduction to make the paper clearer and more readable.
>
>
> [6] Xi, Zerong, and Gita Sukthankar. "A Graph Representation for Autonomous Driving." *The 36th Conference on Neural Information Processing Systems Workshop*. 2022.
>
> [7] [https://nuplan-devkit.readthedocs.io/en/latest/metrics_description.html](https://nuplan-devkit.readthedocs.io/en/latest/metrics_description.html)

---

> ### Author Response · Authors · 2023-11-23
>
> Dear Reviewer LJSU:
>
> As the deadline approaches, we hope that the recent clarifications and revisions to the manuscript have addressed your concerns.  Should you need any further adjustments, we remain ready to continue refining our work.
>
> Best regards,
>
> Authors of Submission 3243

---

### Author Response · Authors · 2023-11-22
**Paper update**

Dear AC and reviewers,

We sincerely appreciate your valuable time and insightful suggestions  to improve the quality and clarity of our manuscript.

Based on the reviewers' comments, we have revised our original manuscript. A summary of the changes is provided below:

- Abstract: Revised last part of abstract to be more precise and concrete.

- Introduction: Adjusted certain broad and complex statements to improve clarity and rigour.

- Memory module: Explained the similarity retrieval process and provided clearer details on memory initialization and storage.

- Comparison with RL method: Added a detailed description of the GRAD method, highlighting its effectiveness in representing driving scenarios and its SOTA performance in the Highway-env.

- Table 1: Added the complete ablation study table to remove any ambiguity.

- Conclusion: Incorporated a part discussing the limitations of our study.

**The changes in the revised manuscript are marked in blue** for easy identification. We hope that our revision makes our work clearer and addresses the concerns of the reviewers.

Best regards,

Authors of Submission 3243

---

### Meta-Review · Area_Chair_fCgM · 2024-01-07

**Metareview:**

The submission proposes a novel system for autonomous driving with large language models. The system is inspired from that many decision making processes of humans are knowledge-driven rather than data-driven. In other words, how can a system utilize existing knowledge for autonomous driving? The submission addressed this question by a system consisting reasoning, memory and reflection models, empowered by LLM. This approach brings extra benefit of generalization and continuous learning. All reviewers also agreed that the proposed system is novel. In addition to suggesting a method from a different perspective, the results are in favor of the proposed method against one of the SOTA methods. Also, the role of each module is fairly well evaluated. Thus, this paper is recommended for accept by all reviewers.

There are two extra suggestions:
* Please add thorough comparison to other SOTA models
* Clarification of the dependency on LLM (and how it was trained)

**Justification For Why Not Higher Score:**

Lack of exhaustive comparison to SOTA models

**Justification For Why Not Lower Score:**

n/a

---

### Decision · Program_Chairs · 2024-01-16

Accept (poster)